# COST-ADAPTIVE RECOURSE RECOMMENDATION BY ADAPTIVE PREFERENCE ELICITATION

## ABSTRACT

Algorithmic recourse recommends a cost-efficient action to a subject to reverse an unfavorable machine learning classification decision. Most existing methods in the literature generate recourse under the assumption of complete knowledge about the cost function. In real-world practice, subjects could have distinct preferences, leading to incomplete information about the underlying cost function of the subject. This paper proposes a two-step approach integrating preference learning into the recourse generation problem. In the first step, we design a question-answering framework to refine the confidence set of the Mahalanobis matrix cost of the subject sequentially. Then, we generate recourse by utilizing two methods: gradient-based and graph-based cost-adaptive recourse that ensures validity while considering the whole confidence set of the cost matrix. The numerical evaluation demonstrates the benefits of our approach over state-of-the-art baselines in delivering cost-efficient recourse recommendations.

## 1 INTRODUCTION

Many machine learning algorithms are deployed to aid significant decisions in various domains. These decisions might have a direct or indirect influence on people's lives, especially in the case of high-profile applications (Verma et al., 2020) such as job hiring (Harris, 2018; Pessach et al., 2020), bank loan (Wang et al., 2020; Turkson et al., 2016) and medical diagnosis (Fatima et al., 2017; Latif et al., 2019). Thus, it's imperative to develop methods to explain the prediction of machine learning models. For instance, when a person applies for a job and is rejected by a predictive model deployed by the employer, the applicant should be notified of the reasoning behind the negative decision and what they could do to be hired.

Recently, algorithmic recourse has become a powerful tool for explaining machine learning (ML) models. Recourse refers to the actions a person should take to achieve an alternative predicted outcome, and it is also known in the literature as a counterfactual explanation. In the case of job hiring, recourse should be individualized suggestions such as "get two more engineering certificates" or "complete one more personal project." When a company suggests a recourse to a subject, this recourse must be valid because the company should accept all applicants who completely implement the suggestions provided in the recommended recourse. Throughout this paper, we use "subject" to refer to the individual who is subject to the prediction of the algorithm. In the context of our job-hiring example, "subject" refers to the job applicant who was rejected by the company.

Several approaches have been proposed to generate recourse for a machine learning model prediction (Karimi et al., 2022; Verma et al., 2020; Stepin et al., 2021). Wachter et al. (2018) used gradient information of the underlying model to generate a counterfactual closest to the input. Ustun et al. (2019) introduced an integer programming problem to find the minimal and actionable change for an input instance. Pawelczyk et al. (2020) leveraged the ideas from manifold learning literature to generate counterfactuals on the high-density data region. Karimi et al. (2020; 2021) generated counterfactual as a sequence of interventions based on a pre-defined causal graph.

These aforementioned approaches all assume that all subjects have the same cost function, for example, the $l_1$ distance (Ustun et al., 2019; Upadhyay et al., 2021; Slack et al., 2021; Ross et al., 2021) or define the same prior causal graph for all subjects (Karimi et al., 2020; 2021). This assumption results in two subjects with identical attributes receiving the same recourse recommendation. Unfortunately, this recourse recommendation is unrealistic in practice because having identical attributes

does not necessarily guarantee that the two subjects will have identical preferences. Indeed, human preferences are strongly affected by many unobservable factors, including historical and societal experiences, which are hardly encoded in the attributes. Thus, the cost functions could be significantly different even between subjects with identical attributes, yet this difference is rarely considered in the recourse generation framework (Yadav et al., 2021).

To mitigate these issues, De Toni et al. (2023) proposed a human-in-the-loop framework to generate a counterfactual explanation uniquely suited to each subject. The proposed method first fixes the initialized causal graph and iteratively learns the subject's specific cost function. Recourse is generated by a reinforcement learning approach that searches for a suitable sequence of interventions. The disadvantage of this approach is that it requires a pre-defined causal graph, which is rarely available in practice (Verma et al., 2020). Besides, Rawal & Lakkaraju (2020) employed the Bradley-Terry model to estimate a universal cost function and then utilized the user input to generate personalized recourse for the user. However, this method is additive in features; therefore, its ability to recover the underlying causal graph remains problematic. Following the same line of work, Yetukuri et al. (2023) captures user preferences via three soft constraints: scoring continuous features, bounding feature values, and ranking categorical features. This method generates recourse via a gradient-based approach. However, the fractional-score concept for user preference might not be as straightforward, especially when the data has many continuous features.

To resolve these problems, we propose a preference elicitation framework that learns the subject's cost function from pairwise comparisons of possible recourses. Compared to De Toni et al. (2023), our framework does not require the causal graph as input, and compared to Rawal & Lakkaraju (2020) and Yetukuri et al. (2023), our framework can perform well even when the dimension of the feature space grows large. This paper contributes by:

- proposing in Section 3 an adaptive preference learning framework to learn the subject's cost function parametrized by the cost matrix of a Mahalanobis distance. This framework initializes with an uninformative confidence set of possible cost matrices. In each round, it determines the next question by finding a pair of recourses corresponding to the most effective cut of the confidence set, that is, a cut that slices the incumbent confidence set most aggressively. The incumbent confidence set shrinks along iterations. We terminate the questioning upon reaching a predefined number of inquiries. The final confidence set is employed for recourse generation.

- proposing in Section 4 two methods for generating recourse under various assumptions of the machine learning models. These methods will consider explicitly the terminal confidence set about the subject's cost matrix. If the model is white-box and differentiable, we can use the cost-adaptive gradient-based recourse-generation method that generates cost-adaptive recourse. Otherwise, we can use the graph-based method to generate the sequential recourse.

Section 5 reports our numerical results. In Appendix A, we also extend our framework to cope with potential inconsistencies in subject responses and extend the heuristics from pairwise comparison to multiple-option questions. All proofs are relegated to the appendix.

**Notations.** Given an integer $d$, we use $\mathbb{S}^d$ and $\mathbb{S}^d_+$ to denote the space of $d$-by-$d$ symmetric matrices and $d$-by-$d$ symmetric positive definite matrices, respectively. The identity matrix is denoted by $I$. The inner product between two matrices $A, B \in \mathbb{S}^d$ is $\langle A, B \rangle = \sum_{i,j} A_{ij} B_{ij}$, and we write $A \preceq B$ to denote that $B - A \in \mathbb{S}^d_+$. The set of integers from 1 to $N$ is $[\![N]\!]$.

## 2 PROBLEM STATEMENT AND SOLUTION OVERVIEW

We are given a binary classifier $\mathcal{C}_\theta : \mathbb{R}^d \to \{0, 1\}$ and access to the training dataset containing $N + M$ instances $x_i \in \mathbb{R}^d$, $i = 1, \ldots, N + M$. The dataset is split into two parts:

- a positive dataset $\mathcal{D}_1 = \{x_1, \ldots, x_N\}$ containing instances with $\mathcal{C}_\theta(x_i) = 1 \; \forall x_i \in \mathcal{D}_1$.
- a negative dataset $\mathcal{D}_0 = \{x_{N+1}, \ldots, x_{N+M}\}$ containing all instances that have the negative predicted outcome, thus $\mathcal{C}_\theta(x_i) = 0 \; \forall x_i \in \mathcal{D}_0$.

Given a subject with input $x_0 \in \mathbb{R}^d$ with a negative predicted outcome $\mathcal{C}_\theta(x_0) = 0$, we make the following assumption on the cost function of this subject.

**Assumption 2.1.** *The subject $x_0$ has a Mahalanobis cost function of the form $c_{A_0}(x, x_0) = (x - x_0)^\top A_0(x - x_0)$ for some symmetric, positive definite matrix $A_0 \in \mathbb{S}_{++}^d$.*

We provide two possible justifications for the aforementioned assumption in Appendix D. First, we describe a sequential control process that affects feature transitions of a subject $x_0$ towards a recourse $x_r$ while minimizing the cost of efforts. We formalize this problem as a Linear Quadratic Regulator, and then we prove that the optimal cost function has the Mahalanobis form, see Section D.1. Second, Appendix D.2 establishes a connection between the linear Gaussian structural causal model and the Mahalanobis cost function. We show that we can recover the Mahalanobis cost preference model with $A_0$ corresponding to the precision matrix of the deviation under linear Gaussian structural equation assumption.

In the above cost function, $A_0$ is the ground-truth matrix specific for subject $x_0$, but it remains elusive to the recourse generation framework. We aim to find $x_r$ which has a positive predicted outcome $\mathcal{C}_\theta(x_r) = 1$ and minimizes the cost $c_{A_0}(x_r, x_0)$. Because the matrix $A_0$ is unknown, we propose an adaptive preference learning approach (Bertsimas & O'Hair, 2013; Vayanos et al., 2020) to approximate the actual cost function $c_{A_0}(x, x_0)$. Our overall approach is as follows: We have a total of $T$ question-answer rounds for cost elicitation. In each round, we choose a pair $(x_i, x_j)$ from the positive dataset $\mathcal{D}_1$. We then ask the subject the following binary question: "Between two possible recourses $x_i$ and $x_j$, which one do you prefer to implement?". The answer from the subject takes one of the three answers: $x_i$ or $x_j$ or indifference. The subject's answer can be used to learn a binary preference relation $\mathcal{P}$. If $x_i$ is preferred to $x_j$, then we denote $x_i \mathcal{P} x_j$; if the subject is indifferent between $x_i$ and $x_j$, then we have simultaneously $x_i \mathcal{P} x_j$ and $x_j \mathcal{P} x_i$. Because both $x_i$ and $x_j$ have positive predicted outcomes, we assume that the subject's preference is solely based on which recourse requires less effort. Assume that $x_i \mathcal{P} x_j$, then $A_0$ should satisfy

$$(x_i - x_0)^\top A_0(x_i - x_0) \le (x_j - x_0)^\top A_0(x_j - x_0). \tag{1}$$

However, to model possible error in the judgment of the subject and to accommodate the indifference answer, we will equip a positive margin $\varepsilon > 0$, and we have $x_i \mathcal{P} x_j$ if and only if:

$$(x_i - x_0)^\top A_0(x_i - x_0) \le (x_j - x_0)^\top A_0(x_j - x_0) + \varepsilon. \tag{2}$$

Let us denote the following matrix $M_{ij} \in \mathbb{S}^d$ as

$$M_{ij} = x_i x_i^\top - x_j x_j^\top + (x_j - x_i)x_0^\top + x_0(x_j - x_i)^\top, \tag{3}$$

then we can rewrite (2) in the form $\langle A_0, M_{ij} \rangle \le \varepsilon$. Let $\mathbb{P}$ be a set of ordered pairs representing the information collected so far about the preference of the subject:

$$\mathbb{P} = \{(i, j) \in [\![N]\!] \times [\![N]\!] \ : \ x_i \mathcal{P} x_j\}.$$

For any preference set $\mathbb{P}$, we can define $\mathcal{U}_\mathbb{P}$ as the set of possible cost matrices $A$ that is consistent with the revealed preferences $\mathbb{P}$:

$$\mathcal{U}_\mathbb{P} \triangleq \{A \in \mathbb{S}_+^d \ : \ \langle A, M_{ij} \rangle \le \varepsilon \ \forall (i, j) \in \mathbb{P}\}, \tag{4}$$

then at any time, we have $A_0 \in \mathcal{U}_\mathbb{P}$. Thus, $\mathcal{U}_\mathbb{P}$ is considered the confidence set of the cost matrix from the viewpoint of the recourse generation framework. Our learning framework aims to reduce the size of $\mathcal{U}_\mathbb{P}$, hoping to pinpoint a small region where $A_0$ may reside. Afterward, we use a recourse generation method adapted to the confidence set $\mathcal{U}_\mathbb{P}$.

We present the overall flow of our framework in Figure 1. In general, our framework addresses several questions of the cost-adaptive recourse-generation approach:

1. What are the questions to ask the subject? If $N$ is large, asking the subject exhaustively for $\mathcal{O}(N^2)$ pairwise comparisons is impossible. Thus, this question aims to find the pair $x_i$ and $x_j$ such that $(i, j) \notin \mathbb{P}$ and $(j, i) \notin \mathbb{P}$, and that adding either one of these two ordered pairs to $\mathbb{P}$ will bring the largest amount of information as possible (in the sense of narrowing down the set $\mathcal{U}_\mathbb{P}$).

2. How to recommend a recourse $x_r$ that minimizes the cost, knowing the confidence set $\mathcal{U}_\mathbb{P}$?

3. What happens if there is inconsistency in the subject's preferences? For example, if there exist three distinct indices $(i, j, k)$ such that the subject states $x_i \mathcal{P} x_j$, $x_j \mathcal{P} x_k$ and $x_k \mathcal{P} x_i$.

The first and third questions are the fundamental questions in preference learning literature (Lu & Shen, 2021; Bertsimas & O'Hair, 2013; Vayanos et al., 2020). In the marketing literature (Toubia et al., 2003; 2004) or recommendation systems literature (Zhao et al., 2016; Rashid et al., 2008; Pu et al., 2012), the preference learning framework aims to recommend products that maximize the utility or preference of subjects. In the adaptive questionnaire framework, we would like to ask questions that give us the most information regardless of the response because the responses to each question are unknown. Moreover, we would like to select the next comparison questions to ask the subject that can maximize the acquired information and reduce the size of the confidence set as quickly as possible (Bertsimas & O'Hair, 2013; Vayanos et al., 2020).

Guided by these ideas, we integrate the adaptive preference learning framework into the recourse generation problem. We show the overall flow of our framework in Figure 1. Our approach generally consists of two phases: preference elicitation and recourse generation. Next, we present the preference elicitation phase in Section 3 and recourse-generation methods in Section 4.

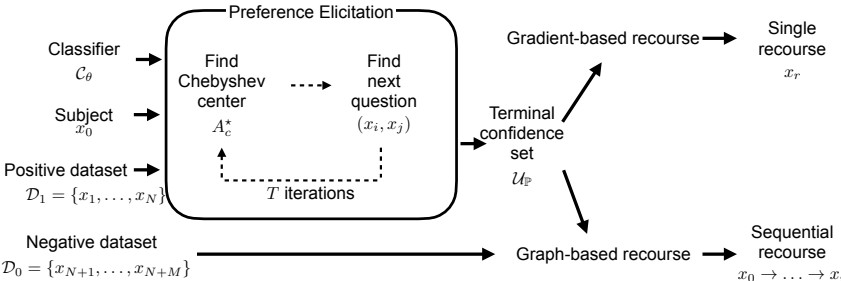

Figure 1: Overall flow of our cost-adaptive recourse recommendation framework. The subject inputs an instance $x_0$. In each of $T$ rounds of question-answer, we first find the Chebyshev center of the set $\mathcal{U}_{\mathbb{P}}$, then select the next question that minimizes the distance to the Chebyshev center. We provide two methods to generate the cost-adaptive recourse: gradient-based and graph-based.

## 3 COST IDENTIFICATION VIA ADAPTIVE PAIRWISE COMPARISONS

### 3.1 FINDING THE CHEBYSHEV CENTER

First, we observe that without any loss of generality, we can impose an upper bound constraint $A \preceq I$ to the set $\mathcal{U}_{\mathbb{P}}$. Indeed, the inequality (1) is invariant with any positive scaling of the matrix $A_0$, and thus, we can normalize $A_0$ so that it has a maximum eigenvalue of one. Adding $A \preceq I$ makes the set $\mathcal{U}_{\mathbb{P}}$ bounded. Given a bounded set $\mathcal{U}_{\mathbb{P}}$, we find the Chebyshev center of $\mathcal{U}_{\mathbb{P}}$ for each question-answer round. Then, we find the question prescribing a hyperplane closest to this center; thus, this hyperplane can be considered the most aggressive cut. Notice that a question involving $x_i$ and $x_j$ can be represented by the hyperplane $\langle A, M_{ij} \rangle = 0$. The confidence set $\mathcal{U}_{\mathbb{P}}$ is simply a polytope in the space of positive definite matrices.

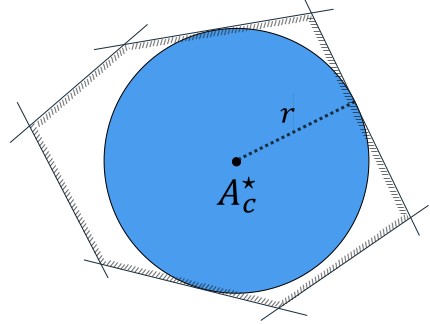

Figure 2: Illustration of the Chebyshev center. Black lines represent the hyperplanes $\langle A, M_{ij} \rangle = \varepsilon$ for $(i, j) \in \mathbb{P}$ defining the boundaries of the polytope $\mathcal{U}_{\mathbb{P}}$. The ball centered at the Chebyshev center $A_c^{\star}$ with radius $r$ is the largest inscribed ball of $\mathcal{U}_{\mathbb{P}}$.

We first consider finding the Chebyshev center of the set $\mathcal{U}_{\mathbb{P}}$. For any bounded set with a non-empty interior, the Chebyshev center is the center of a ball with the largest radius inside the set. Thus, given a confidence set $\mathcal{U}_{\mathbb{P}}$, its Chebyshev center represents a safe point estimate of the true cost matrix. The Chebyshev center $A_c^{\star}$ and its corresponding radius $r^{\star}$ of $\mathcal{U}_{\mathbb{P}}$ is the optimal solution of the problem

$$(A_c^{\star}, r^{\star}) = \arg \max_{A_c \in \mathbb{S}_+^d, \, r \in \mathbb{R}_+} \left\{ r \, : \, \|A - A_c\|_F^2 \leq r^2 \,\, \forall A \in \mathcal{U}_{\mathbb{P}} \right\}.$$

For our problem, the Chebyshev center can be found by solving a semidefinite program resulting from the following theorem.

**Theorem 3.1** (Chebyshev center). *Suppose that $\mathcal{U}_{\mathbb{P}}$ has a non-empty interior. The Chebyshev center $A_c^\star$ of the set $\mathcal{U}_{\mathbb{P}}$ can be found by solving the following semidefinite program*

$$
\begin{aligned}
\max \quad & r \\
\mathrm{s.\,t.} \quad & A_c \in \mathbb{S}_+^d, \; r \in \mathbb{R}_+ \\
& A_c \preceq I, \quad \langle A_c, M_{ij} \rangle + r\|M_{ij}\|_F \leq \varepsilon \quad \forall (i,j) \in \mathbb{P}.
\end{aligned}
\tag{5}
$$

### 3.2  Recourse-Pair Determination

Finding the next question to ask the subject is equivalent to finding two indices $(i,j) \in [\![N]\!] \times [\![N]\!]$, corresponding to two recourses $x_i$ and $x_j$ in the positive dataset $\mathcal{D}_1$, such that the corresponding hyperplane $\langle A, M_{ij} \rangle = 0$ is as close to the Chebyshev center $A_c^\star$ as possible. This is equivalent to solving the minimization problem

$$
\min_{(i,j) \in [\![N]\!] \times [\![N]\!]} \frac{|\langle A_c^\star, M_{ij} \rangle|}{\|M_{ij}\|_F},
$$

where the matrix $M_{ij}$ is calculated as in (3). The objective function of the above problem is simply the projection distance of $A_c^\star$ to $\langle A, M_{ij} \rangle = 0$ under the Frobenius norm.

**Similar cost heuristics.** An exhaustive search over all pairs of indices $(i,j)$ requires an $\mathcal{O}(N^2)$ complexity. This search may become too expensive for large datasets because we must conduct one separate search at each round. We propose a heuristic that can produce reasonable questions in a limited time to alleviate this burden. This heuristics is based on the following observation: given an incumbent Chebyshev center $A_c^\star$, two valid recourses $x_i$ and $x_j$ are more comparable to each other if their resulting costs measured with respect to $A_c^\star$ are close to each other, that is, $c_{A_c^\star}(x_i, x_0) \approx c_{A_c^\star}(x_j, x_0)$. If their costs are too different, for example, $c_{A_c^\star}(x_i, x_0) \ll c_{A_c^\star}(x_j, x_0)$, then it is highly likely that the subject will prefer $x_i$ to $x_j$ uniformly over the set of possible weighting matrices in $\mathcal{U}_{\mathbb{P}}$. Profiting from this observation, we consider the following similar-cost heuristic:

- Step 1: Compute the distances from $x_i$ to $x_0$: $s_i = (x_i - x_0)^\top A_c^\star (x_i - x_0)$ for all $i \in [\![N]\!]$,
- Step 2: Sort $s_i$ in a non-decreasing order. The sorted vector is denoted by $(s_{[1]}, \ldots, s_{[N]})$,
- Step 3: For each $i = 1, \ldots, N-1$, choose a pair of adjacent cost samples $x_{[i]}$ and $x_{[i+1]}$ corresponding to $s_{[i]}$ and $s_{[i+1]}$, then compute the projection distance of the incumbent center $A_c^\star$ to the hyperplane $\langle M_{[i],[i+1]}, A \rangle = 0$.
- Step 4: Pick a pair of $([i], [i+1])$ that induces the smallest projection distance in Step 3.

In step 2, sorting costs $\mathcal{O}(N \log N)$. Nevertheless, in Step 3, we only need to compute $N$ times the projection distance by looking at pairs of adjacent costs, contrary to the total number of $\mathcal{O}(N^2)$ pairs. The comparison between similar cost heuristics and exhaustive search is relegated to Appendix B.

## 4  Cost-Adaptive Recourse Recommendation

Given the subject input $x_0$, this section explores two generalizations to generate single and sequential recourses, adapted to the terminal confidence set $\mathcal{U}_{\mathbb{P}}$ of the cost metric. In Section 4.1, we generalize the gradient-based counterfactual generation method in Wachter et al. (2018). In Section 4.2, we generalize the graph-based counterfactual generation method in Poyiadzi et al. (2020).

### 4.1  Gradient-based Cost-adaptive Single Recourse

Given a machine learning model $f_\theta : \mathbb{R}^d \to (0,1)$ that outputs the probability of being classified in the favorable group. The binary classifier $\mathcal{C}_\theta : \mathbb{R}^d \to \{0,1\}$ takes the form of a threshold policy

$$
\mathcal{C}_\theta(x) = \begin{cases} 1 & \text{if } f_\theta(x) \geq 0.5, \\ 0 & \text{otherwise,} \end{cases}
$$

where we have used a threshold of 0.5 similar to the setting in Wachter et al. (2018).

We suppose that we have access to the probability output $f_\theta$. Let $l$ be a differentiable loss function that minimizes the gap between $f_\theta(x)$ and the decision threshold 0.5; one can think of $l(f_\theta(x), 1)$ as the term that promotes the validity of the recourse. Given a weight $\lambda \geq 0$ which balances the trade-off between the validity and the (worst-case) cost, we can generate a recourse for an input instance $x_0$ by solving

$$\min_{x \in \mathcal{X}} \left\{ l(f_\theta(x), 1) + \lambda \max_{A \in \mathcal{U}_\mathbb{P}} (x - x_0)^\top A (x - x_0) \right\}. \tag{6}$$

A practical choice for loss function is the quadratic loss $l(f_\theta(x), 1) = (f_\theta(x) - 0.5)^2$, which is a differentiable function in $x$ (Wachter et al., 2018). Under a mild condition about the uniqueness of the optimal solution to the inner maximization problem, the cost term in the objective of (6) is also differentiable. Thus, one can invoke a (projected) gradient descent algorithm to solve (6) and find the recourse. Algorithm 1 proceeds iteratively to solve problem (6). In each iteration, we first find a matrix $A^\star$ of the max problem with a solver such as Mosek (MOSEK ApS, 2019), and then we take a gradient step in the variable $x$ using the computed gradient. The next incumbent solution is the projection onto the set $\mathcal{X}$, where $\Pi_\mathcal{X}$ denotes the projection onto $\mathcal{X}$. Furthermore, similar to Wachter et al. (2018), we can add an early stopping criterion for Algorithm 1. For example, we can stop the algorithm at iteration $t$ if $\mathcal{C}_\theta(x^t) = 1$.

---

**Algorithm 1** Gradient descent algorithm for cost-adaptive recourse generation

**Input:** Input $x_0$ s.t. $\mathcal{C}_\theta(x_0) = 0$
**Parameters:** $\lambda > 0$, learning rate $\alpha$
**Initialization:** Set $x^0 \leftarrow x_0$
**for** $t = 0, \ldots, T - 1$ **do**

$$A^\star \leftarrow \max_{A \in \mathcal{U}_\mathbb{P}} (x^t - x_0)^\top A (x^t - x_0)$$

$$g \leftarrow \nabla l(f_\theta(x^t), 1) + 2\lambda A^\star (x^t - x_0)$$

$$x^{t+1} \leftarrow \Pi_\mathcal{X}(x^t - \alpha g),$$

**end for**
**Output:** $x^T$

---

## 4.2 GRAPH-BASED COST-ADAPTIVE SEQUENTIAL RECOURSE

In Section 4.1, we introduce a gradient-based recourse-generation method. However, this approach requires access to the gradient information, which is restricted in some real-world applications (Ilyas et al., 2018; Alzantot et al., 2019). In this section, we present a model-agnostic recourse-generation approach that leverages the ideas from FACE (Poyiadzi et al., 2020). After $T$ rounds of questions in Section 3, we solve problem (5) to find the Chebyshev center $A^\star$ of the terminal confidence set $\mathcal{U}_\mathbb{P}$.

**Graph construction.** We first build a directed graph $\mathcal{G} = (\mathcal{V}, \mathcal{E})$ that represents the geometry of the available data: each node $x_i \in \mathcal{V} = \{x_0\} \cup \mathcal{D}_1 \cup \mathcal{D}_0$ corresponds to a data sample, and an edge $(x_i, x_j) \in \mathcal{E}$ represents a feasible transition from node $x_i$ to node $x_j$. We compute the edge weight $w_{ij} = c_{A^\star}(x_i, x_j)$ based on Mahalanobis cost function associated with matrix $A^\star$. Finally, $w_{ij} = \infty$ for $(x_i, x_j) \notin \mathcal{E}$.

**Sequential recourse generation.** Recall that $\mathcal{D}_1$ is the set of all vertices with favorable predicted outcomes. A

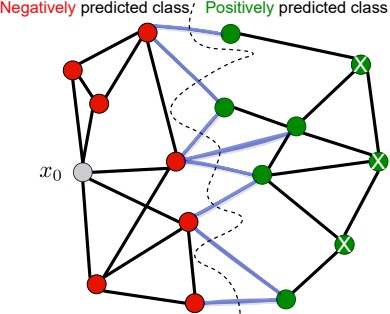

Figure 3: The illustration of $\mathcal{G}$, showing negatively predicted samples as red circles and positively predicted samples as green circles. The input instance $x_0$ is a gray circle. The terminal edges and unreachable nodes of flows in $\mathcal{F}$ are blue edges and green nodes with white crosses, respectively.

one-step recourse recommendation suggests a single continuous action from $x_0$ to $x_r$ (e.g., Ustun et al., 2019; Mothilal et al., 2020). A sequential recourse is a directed path from the input instance $x_0$ to a node $x_r \in \mathcal{D}_1$; each transition in the path is a concrete action that the subject has to implement to move towards $x_r$. A sequential recourse has several advantages compared to the one-step ones: plausibility and sparsity. In real-world applications, sequential steps are more plausible than a one-step continuous change (Ramakrishnan et al., 2020; Singh et al., 2021). Moreover, recent work shows that sequential recourse promotes sparsity, allowing subjects to modify a few features at each step (Verma et al., 2022). For illustration purposes, we present an example of sequential recourse in Appendix B. The cost of a sequential recourse is computed by the sum of all the edge weights in the

path. Thus, we can recommend a sequential and actionable recourse by finding a path that originates from $x_0$ and ends at the node $x_r^\star \in \mathcal{D}_1$ with the lowest path cost.

**Worst-case sequential recourse generation**. After conducting $T$ rounds of questioning in Section 3, we obtain the confidence set $\mathcal{U}_\mathbb{P}$ for the parameter $A_0$. However, the precise value of $A_0$ remains unknown. In this section, we focus on minimizing the total cost of the sequential recourse subject to the most unfavorable scenario of $A_0$ within the final confidence set.

Let $\mathcal{F}$ denote the set containing all possible flows from the input subject $x_0$ to a node in $\mathcal{D}_1$. Mathematically, we can write $\mathcal{F}$ as

$$\mathcal{F} = \left\{ f_{ij} \in \{0,1\} \ \forall (x_i, x_j) \in \mathcal{E} : \begin{array}{l} \sum_{(x_0, x_j) \in \mathcal{E}} z_{0j} - \sum_{(x_j, x_0) \in \mathcal{E}} z_{j0} = 1 \\ \sum_{x_j \in \mathcal{D}_1, x_i \in \mathcal{V} \backslash \mathcal{D}_1} z_{ij} = 1 \\ \sum_{(x_i, x_j) \in \mathcal{E}} z_{ij} - \sum_{(x_j, x_i) \in \mathcal{E}} z_{ji} = 0 \ \ \forall x_i \in \mathcal{V} \backslash \mathcal{D}_1, x_i \neq x_0 \end{array} \right\}.$$

Figure 3 illustrates the visual representation of the set $\mathcal{F}$. The first constraint ensures that the total flow out of $x_0$ is precisely one. The second constraint enforces the terminal condition for flows, halting the flow once it reaches the first node in the positive class. In the visual depiction in Figure 3, the terminal edges of flows are visually distinguished as blue edges. Consequently, positive nodes without direct connections from negative nodes are not part of any flows, and they are identifiable as green nodes with white crosses in Figure 3. The third constraint imposes flow conservation at each negatively predicted node. For any $f \in \mathcal{F}$, we have $f_{ij} = 1$ if the edge $(x_i, x_j)$ constitutes one (actionable) step in the path. The optimal cost-robust sequential recourse is defined to be the optimal flow of the min-max problem

$$\min_{f \in \mathcal{F}} \ \max_{A \in \mathcal{U}_\mathbb{P}} \sum_{(x_i, x_j) \in \mathcal{E}} w_{ij}(A) f_{ij}, \tag{7}$$

with the edge weight depends explicitly on the weighting matrix $A$ as $w_{ij}(A) = (x_i - x_j)^\top A (x_i - x_j)$. The next proposition asserts an equivalent form of (7) as a single-layer minimization problem.

**Proposition 4.1** (Equivalent formulation). *Problem* (7) *is equivalent to*

$$\begin{cases} \min & \langle U, I \rangle + \varepsilon \sum_{(i,j) \in \mathbb{P}} t_{ij} \\ \text{s.t.} & f \in \mathcal{F}, \ t_{ij} \geq 0 \ \forall (i,j) \in \mathbb{P}, \ U \in \mathbb{S}_+^d \\ & U + \sum_{(i,j) \in \mathbb{P}} M_{ij} t_{ij} \succeq \sum_{(x_i, x_j) \in \mathcal{E}} (x_i - x_j)(x_i - x_j)^\top f_{ij}. \end{cases} \tag{8}$$

Problem (8) is a binary semidefinite programming problem, which is challenging to solve due to its combinatorial nature. Consequently, finding an optimal sequential recourse can be a daunting task. To address this issue, we propose an alternative approach. Specifically, we associate the weight of each edge $(x_i, x_j)$ with its maximum cost taken over all possible values of $A$ in the set $\mathcal{U}_\mathbb{P}$:

$$\bar{w}_{ij} = \max_{A \in \mathcal{U}_\mathbb{P}} w_{ij}(A) = \begin{cases} \max & \langle A, (x_i - x_j)(x_i - x_j)^\top \rangle \\ \text{s.t.} & 0 \preceq A \preceq I, \ \ \langle A, M_{i'j'} \rangle \leq \varepsilon \ \forall (i', j') \in \mathbb{P}. \end{cases}$$

Given a graph $\mathcal{G}$ with the worst-case weight matrix $[\bar{w}_{ij}]$, we find the shortest paths from $x_0$ to each positively-predicted node in $\mathcal{D}_1$. The recommended sequential recourse is the path that originates from $x_0$ and ends at the node $x_r^\star \in \mathcal{D}_1$ with the lowest path cost.

## 5 NUMERICAL EXPERIMENTS

We evaluate our method, Cost-Adaptive Recourse Recommendation by Adaptive Preference Elicitation (ReAP), using synthetic data and seven real-world datasets: German, Bank, Student, Adult, COMPAS, GMC, and HELOC. Notably, these datasets are commonly used in recourse literature (Verma et al., 2020; Upadhyay et al., 2021; Mothilal et al., 2020). In the main paper, we present the results for Synthesis, German, Bank, and Student datasets. The results for other datasets can be found in the appendix. We compare our approach against the recourse-generation baselines implemented in CARLA Pawelczyk et al. (2021). For the gradient-based single recourse method in Section 4.1, we compare our method to Wachter Wachter et al. (2018) and DiCE Mothilal et al. (2020). For the graph-based sequential recourse method in Section 4.2, we compare our method to FACE Poyiadzi et al. (2020). Codes for the experiments in the main paper are provided in the supplementary material. In Appendix B, we present the detailed implementation and numerical results for additional datasets, providing a benchmarking performance for the proposed heuristics and an additional comparison against PEAR (De Toni et al., 2023).

### 5.1 EXPERIMENTAL SETUP

**Data preprocessing.** Following Mothilal et al. (2020), we preprocess the data using the min-max standardizer for continuous features and one-hot encoding for categorical features.

**Classifier.** For each dataset, we perform an 80-20 uniformly split (80% for training) of the original dataset. Then we train an MLP classifier $\mathcal{C}_\theta$ on the training data. We use the test data to benchmark the performance of different recourse-generation methods.

**Cost matrix generation.** We generate 10 ground-truth matrices $A_0$ with this procedure: first, we generate a matrix $A \in \mathbb{R}^{d \times d}$ of random, standard Gaussian elements, where $d$ is the dimension of $x_0$. Then we compute $A_0 = AA^\top$ and normalize $A_0$ to have a unit spectral radius by taking $A_0 \leftarrow A_0/\sigma_{\max}(A_0)$, where $\sigma_{\max}$ is the maximum eigenvalue function.

For an input $x_0$ and a ground-truth matrix $A_0$, we choose $T$ questions using the similar-cost heuristics in Section 3.2 to find the set $\mathcal{U}_\mathbb{P}$. After $T$ rounds of question-answers, we solve (5) using MOSEK to find the Chebyshev center $A^\star$ of the terminal confidence set $\mathcal{U}_\mathbb{P}$. Then, we generate recourse using the gradient-based method in Section 4.1 and the graph-based method in Section 4.2. Note that with $T = 0$, we haven't asked any questions. Thus, $A^\star = \frac{1}{2}I$ (an uninformative estimate). Hence, all algorithms share the same cost function. Within this context, the proposed worst-case sequential recourse generation in Section 4.2 demonstrates the effectiveness as it manages to provide an acceptable recourse for challenging scenarios within the domain where $A$ is a matrix satisfying $A \preceq I, A \in \mathbb{S}_+^d$. This approach also proves valuable when users' responses contain significant noise and inconsistencies, resulting in a still large search space for $A_0$ in the final round.

### 5.2 METRICS FOR COMPARISON

We compare different recourse-generation methods using the following metrics:

**Validity.** A recourse $x_r$ generated by a recourse-generation method is valid if $\mathcal{C}_\theta(x_r) = 1$. We compute validity as the fraction of instances for which the recommended recourse is valid.

**Cost.** For the gradient-based single recourse method, we calculate the cost of a recourse $x_r$ as the Mahalanobis distance between $x_r$ and $x_0$ evaluated with the ground-truth matrix $A_0$ as $c_{A_0}(x_r, x_0)$.

**Shortest-path cost.** For the graph-based recourse-generation, we report the cost of a sequential recourse $x_0 \to \ldots \to x_r$ as the path cost from input $x_0$ to $x_r$, evaluated with $A_0$.

**Mean rank.** We borrow the ideas from Bertsimas & O'Hair (2013) and consider the mean rank metric for ranking recourses based on subject preference. We first rank all of the recourses in the positive dataset $\mathcal{D}_1$ with their preferences according to the ground-truth matrix $A_0$. Thus, the recourse with the smallest cost is ranked 1, and the recourse with the largest cost is ranked $N$ ($N$ is the total number of recourses in the positive dataset). We then find the top $K$ recourses according to the cost metric $c_{A^\star}(x, x_0)$ and compare the selected solutions with the true rank of the recourse. Therefore, smaller values indicate that the matrix $A^\star$, the Chebyshev center of the terminal confidence set, is closer to the ground truth $A_0$. Each recourse $x_i \in \mathcal{D}_1$ thus can be assigned with a rank $r_i \in [1, \ldots, N]$. We compute the normalized mean rank of top $K$ recourses as $r_{\mathrm{mean}} = (\sum_{i=1}^K r_i - r_{\min})/r_{\max}$ where $r_{\min} = \sum_{i=1}^K i = (K+1)K/2$ and $r_{\max} = \sum_{i=N-K+1}^N i = (2N - K + 1)K/2$ are normalizing constants so that $r_{\mathrm{mean}} \in (0, 1)$.

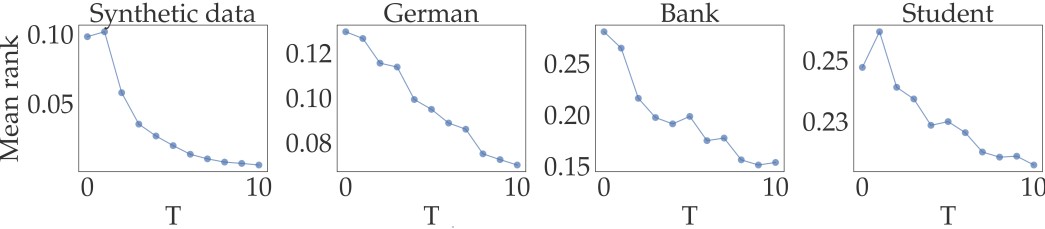

Figure 4: Impact of the number of questions $T$ to the average mean rank on synthetic data and three real-world datasets. As the number of questions increases, the mean rank tends to decrease, highlighting that the Chebyshev center tends closer to the ground truth $A_0$.

### 5.3 NUMERICAL RESULTS

We conduct three experiments to study the efficiency of our framework in generating cost-adaptive recourses. First, we study the impact of the number of questions $T$ on the mean rank. Then, we compare our two cost-adaptive recourse-generation methods: gradient-based and graph-based, with the recourse-generation baselines implemented in CARLA (Pawelczyk et al., 2021). Appendix B provides additional numerical results and discussions.

**Impact of the number of questions $T$ to the mean rank.** Here, we analyze the impact of the number of questions $T$ on the mean rank. We first fix the parameter $\varepsilon$ and vary the number of questions $T \in [0, 10]$. For each value of $T$, we choose $T$ questions with the heuristics in Section 3.2 and solve problem (5) to find the center $A^\star$. Then, we evaluate the mean rank with $A^\star$. Figure 4 demonstrates that the average mean rank decreases as the number of questions increases. This implies that the Chebyshev center $A^\star$ comes closer to the ground truth $A_0$ with the more questions we ask, leading to a more accurate estimate of the actual cost function.

**Gradient-based cost-adaptive recourse.** In this experiment, we generate recourse using our gradient-based recourse-generation method. We compute the cost as the Mahalanobis distance described in Section 5.2. We compare our method with three baselines: Wachter and DiCE. Table 1 demonstrates that DiCE has the highest cost across all datasets, and its validity isn't perfect in the German, Bank, and Student datasets. Our method has similar validity to Wachter but at a lower cost in three out of four datasets. It's important to note that if $T = 0$, the Chebyshev center is $A^\star = \frac{1}{2}I$, and the cost metric $c_{A^\star}(x, x_0)$ becomes the squared Euclidean distance between $x$ and $x_0$, which DiCE and Wachter directly optimize. Thus, these results indicate that our approach effectively adjusts to the subject's cost function and adequately reflects the individual subject's preferences.

Table 1: Benchmark of Cost and Validity between gradient-based methods on four datasets.

| Dataset | Methods | Cost | Validity |
|---------|---------|------|----------|
| Synthetic | DiCE | $0.31 \pm 0.27$ | $\mathbf{1.00} \pm 0.00$ |
| | Wachter | $0.12 \pm 0.14$ | $\mathbf{1.00} \pm 0.00$ |
| | ReAP | $\mathbf{0.10} \pm 0.15$ | $\mathbf{1.00} \pm 0.00$ |
| German | DiCE | $0.10 \pm 0.37$ | $0.96 \pm 0.19$ |
| | Wachter | $0.03 \pm 0.02$ | $\mathbf{1.00} \pm 0.00$ |
| | ReAP | $\mathbf{0.01} \pm 0.01$ | $\mathbf{1.00} \pm 0.00$ |
| Bank | DiCE | $1.43 \pm 0.61$ | $0.99 \pm 0.10$ |
| | Wachter | $0.11 \pm 0.10$ | $\mathbf{1.00} \pm 0.00$ |
| | ReAP | $\mathbf{0.08} \pm 0.08$ | $\mathbf{1.00} \pm 0.00$ |
| Student | DiCE | $0.07 \pm 0.18$ | $0.64 \pm 0.48$ |
| | Wachter | $\mathbf{0.05} \pm 0.07$ | $\mathbf{1.00} \pm 0.00$ |
| | ReAP | $\mathbf{0.05} \pm 0.07$ | $\mathbf{1.00} \pm 0.00$ |

**Graph-based cost-adaptive recourse.** In this experiment, we generate recourse using the graph-based sequential recourse method. We compute the cost of a sequential recourse as the shortest-path cost described in Section 5.2. We compare our graph-based method with FACE. Table 2 demonstrates that our ReAP framework has the lowest cost across all four datasets. The validity of the two methods is perfect in all four datasets because the two methods both find a path from the input node $x_0$ to the node $x_r \in \mathcal{D}_1$. As mentioned above, if $T = 0$, the cost metric $c_{A^\star}(x, x_0)$ becomes squared of the Euclidean distance between $x$ and $x_0$, and FACE builds the graph using this Euclidean metric. These observations show that our graph-based method accurately captures the subjects' preferences and adapts to their cost function.

Table 2: Benchmark of Path cost between graph-based ReAP and FACE. All methods attain the validity of $1.00 \pm 0.00$.

| Dataset | Methods | Path cost |
|---------|---------|-----------|
| Synthetic | FACE | $0.73 \pm 0.55$ |
| | ReAP | $\mathbf{0.70} \pm 0.56$ |
| German | FACE | $0.66 \pm 0.48$ |
| | ReAP | $\mathbf{0.53} \pm 0.49$ |
| Bank | FACE | $1.20 \pm 0.69$ |
| | ReAP | $\mathbf{0.82} \pm 0.39$ |
| Student | FACE | $1.10 \pm 0.76$ |
| | ReAP | $\mathbf{1.04} \pm 0.66$ |

## 6 CONCLUSIONS

This work proposes an adaptive preference learning framework for the recourse generation problem. Our proposed framework aims to approximate the true cost matrix of the subject in an iterative manner using a few rounds of question-answers. At each round, we select the question corresponding to the most effective cut of the confidence set of possible cost matrices. We provide two recourse-generation methods: gradient-based and graph-based cost-adaptive recourse. Finally, we generalize our framework to handle inconsistencies in subject responses and extend the heuristics to choose the questions from pairwise comparison to multiple-option questions. Extensive numerical experiments show that our framework can adapt to the subject's cost function.

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

**Broader Impacts and Limitations.** This paper aims to generate recourse adapted to each subject's cost function. The gradient-based method in Section 4.1 and graph-based method in Section 4.2 require access to gradient information and training data, respectively. We want to highlight that access to this information is leveraged in existing gradient-based methods such as ROAR (Upadhyay et al., 2021) or graph-based methods such as FACE (Poyiadzi et al., 2020). A frequent criticism of the needed access to data or model information is that it could violate the privacy of the machine learning system. Moreover, recent research demonstrates that solutions produced by recourse-generation methods and those produced by adversarial example-generating algorithms are highly comparable (Pawelczyk et al., 2022). To increase the system's trustworthiness, a decision-making system must, therefore, be able to discern between an adversarial example and a recourse. We may use various strategies and approaches to ensure privacy to overcome these problems. However, these issues are outside our work's scope, so we left these problems for future research.

## A    GENERALIZATIONS

In this section, we describe two main generalizations of our framework: Section A.1 considers possible inconsistencies in the preference elicitation of the subject, and Section A.2 considers the generalization to a $k$-way questioning.

### A.1    ADDRESSING INCONSISTENCY IN COST ELICITATION

It is well-documented that human responses in behavior elicitation may exhibit inconsistencies. Inconsistencies occur when there exist three distinct indices $(i, j, k)$ such that the user states $x_i \mathcal{P} x_j$, $x_j \mathcal{P} x_k$ and $x_k \mathcal{P} x_i$. In this case, the set $\mathcal{U}_\mathbb{P}$ becomes empty, and finding a Chebyshev center $A_c^\star$ is impossible. One practical approach to alleviate the effect of the inconsistency is to allow a fraction of the stated preferences to be violated in the definition of the cost-uncertainty set $\mathcal{U}_\mathbb{P}$. Let $|\mathbb{P}|$ denote the cardinality of the set $\mathbb{P}$. Suppose we tolerate $\alpha\%$ of inconsistency, i.e., there are at most $\alpha|\mathbb{P}|$ preferences in the set $\mathbb{P}$ that can be violated. We define $\mathcal{U}_\mathbb{P}^\alpha$ as the set of possible cost matrices $A$ with at most $\alpha\%$ inconsistency with the preference set $\mathbb{P}$. This set can be represented using auxiliary binary variables as

$$\mathcal{U}_\mathbb{P}^\alpha = \left\{ A \in \mathbb{S}_+^d : \begin{array}{l} \exists \gamma_{ij} \in \{0, 1\} \ \forall (i, j) \in \mathbb{P} \\ \sum_{(i,j) \in \mathbb{P}} \gamma_{ij} \leq \alpha|\mathbb{P}| \\ \langle A, M_{ij} \rangle \leq \varepsilon + \gamma_{ij} \mathbb{M} \end{array} \right\}, \tag{9}$$

where $\mathbb{M}$ is a big-M constant. Intuitively, $\gamma_{ij}$ is an indicator variable: $\gamma_{ij} = 1$ implies that the preference $x_i \mathcal{P} x_j$ is inconsistent, and thus the corresponding halfspace becomes $\langle A, M_{ij} \rangle \leq \varepsilon + \mathbb{M}$, which is a redundant constraint. The Chebyshev center of the set $\mathcal{U}_\mathbb{P}^\alpha$ can be found by solving a binary semidefinite program.

**Theorem A.1** (Chebyshev center with inconsistent elicitation). *Given $\alpha \in (0, 1)$. The Chebyshev center $A_c^\star$ of the set $\mathcal{U}_\mathbb{P}^\alpha$ can be found by solving the binary semidefinite program*

$$\begin{array}{ll} \max & r \\ \text{s.t.} & A_c \in \mathbb{S}_+^d, \ r \in \mathbb{R}_+, \ \gamma_{ij} \in \{0, 1\} \quad \forall (i, j) \in \mathbb{P} \\ & \langle A_c, M_{ij} \rangle + r\|M_{ij}\|_F \leq \varepsilon + \gamma_{ij} \mathbb{M} \quad \forall (i, j) \in \mathbb{P} \\ & \sum_{(i,j) \in \mathbb{P}} \gamma_{ij} \leq \alpha|\mathbb{P}| \\ & A_c \preceq I, \end{array} \tag{10}$$

*where $\mathbb{M}$ is a big-M constant.*

*Proof of Theorem A.1.* Using the definition of the set $\mathcal{U}_\mathbb{P}$ as in (9), the optimization problem to find the Chebyshev center and its radius can be rewritten as

$$\begin{array}{ll} \max & r \\ \text{s.t.} & A_c \in \mathbb{S}_+^d, \ r \in \mathbb{R}_+ \\ & \langle A_c + \Delta, M_{ij} \rangle \leq \varepsilon \ \forall \Delta \in \mathcal{B}_r, \ \forall (i, j) \in \mathbb{P} \\ & \sum_{(i,j) \in \mathbb{P}} \gamma_{ij} \leq \alpha|\mathbb{P}|. \end{array}$$

where $\mathcal{B}_r$ is a ball of symmetric matrices with Frobenius norm bounded by $r$:

$$\mathcal{B}_r = \{\Delta \in \mathbb{S}^d : \|\Delta\|_F \leq r\}.$$

Pick any $(i, j) \in \mathbb{P}$, the semi-infinite constraint

$$\langle A_c + \Delta, M_{ij} \rangle \le \varepsilon + \gamma_{ij} \mathbb{M} \ \forall \Delta \in \mathcal{B}_r$$

is equivalent to the robust constraint

$$\langle A_c, M_{ij} \rangle + \sup_{\|\Delta\|_F \le r} \langle \Delta, M_{ij} \rangle \le \varepsilon + \gamma_{ij} \mathbb{M}.$$

Because the Frobenius norm is a self-dual norm, we have

$$\sup_{\|\Delta\|_F \le r} \langle \Delta, M_{ij} \rangle = r \|M_{ij}\|_F.$$

Replacing the above equation to the optimization problem completes the proof. $\qquad \square$

Unfortunately, problem (10) is a binary SDP, and state-of-the-art solvers such as Mosek and GUROBI do not support this class of problem. Adhoc methods to solve binary SDP can be found in Ni & So (2018) and the references therein.

### A.2 Multiple-option Questions

Previous results rely on the pairwise comparison settings: given two valid recourses, $x_i$ and $x_j$, the subject indicates one preferred option. These settings can be easily generalized to $k$-option comparison: Given $k$ distinct indices $i_1, \ldots, i_k$, the subject is asked "Which recourse among $x_{i_1}, \ldots, x_{i_k}$ do you prefer the most?." The answer from the subject will reveal $k - 1$ binary preferences: for example, if the subject prefers $x_{i_1}$ the most, then it is equivalent to a revelation of $k - 1$ preferences: $x_{i_1} \mathcal{P} x_{i_2}, \ldots, x_{i_1} \mathcal{P} x_{i_k}$. Thus, if we use a $k$-option question, we can add $k - 1$ relations to the set $\mathbb{P}$, which correspond to $k - 1$ hyperplanes to the set $\mathcal{U}_{\mathbb{P}}$. The computation of the Chebyshev center $A_c^\star$ in Section 3.1 remains invariant. The only added complication is the increased complexity in searching for the next $k$ recourses to ask the subject: instead of $\mathcal{O}(N^2)$ questions, the space of possible questions is now of order $N^k$. Fortunately, we can slightly modify the similar cost heuristics to accommodate the $k$-option questions. More specifically, in Step 3 of the heuristics, we can use the following:

- Step 3: For each $i = 1, \ldots, N - k + 1$, choose a tuple of adjacent cost samples $(x_{[i]}, \ldots, x_{[i+k-1]})$ corresponding to $k$-adjacent costs $(s_{[i]}, \ldots, s_{[i+k-1]})$, then compute the *average* projection distance of the incumbent center $A_c^\star$ to the hyperplanes $\langle M_{[i+k'],[i+k'+1]}, A \rangle = 0$ for $k' = 0, \ldots, k - 2$.

The complexity of this heuristics remains $\mathcal{O}(N \log(N))$.

## B Additional Experiments

In this section, we provide the detailed implementation and additional numerical results. All codes can be accessed from `https://anonymous.4open.science/r/ReAP-07E9/`.

### B.1 Datasets

**Real-world data.** We use seven real-world datasets which are popular in the settings of recourse-generation (Mothilal et al., 2020; Upadhyay et al., 2021): German credit (Dua & Graff, 2017), Bank (Dua & Graff, 2017), Student performance (Cortez & Silva, 2008), Adult (Becker & Kohavi, 1996), COMPAS, GMC and HELOC (Pawelczyk et al., 2021). We describe the selected subset of features from German, Bank, and Student datasets in Table 3. Additionally, we follow the same features selection procedure for Adult, COMPAS Recidivism Racial Bias, Give Me Some Credit (GMC), and HELOC datasets as in Pawelczyk et al. (2021).

**Synthetic data.** Following previous work (Nguyen et al., 2022), we generate the synthetic dataset with two-dimensional data samples by sampling uniformly in a rectangle $x = (x_1, x_2) \in [-2, 4] \times [-2, 7]$ with the following labeling function $f$:

$$f(x) = \begin{cases} 1 & \text{if } x_2 \ge 1 + x_1 + 2x_1^2 + x_1^3 - x_1^4, \\ 0 & \text{otherwise.} \end{cases}$$

Table 3: Features selection from German, Bank, and Student datasets in our experiments.

| Dataset | Features |
|---|---|
| German | Status, Duration, Credit amount, Personal Status, Age |
| Bank | Age, Education, Balance, Housing, Loan, Campaign, Previous, Outcome |
| Student | Age, Study time, Famsup, Higher, Internet, Health, Absences, G1, G2 |

**Example of gradient-based single recourse and graph-based sequential recourse.** We provide an example of a gradient-based and a graph-based recourse on the Bank dataset in Figure 5. A one-step recourse specifies only the final state at which the model yields a favorable outcome. A sequential recourse consists of several smaller steps that lead the subject toward a favorable outcome.

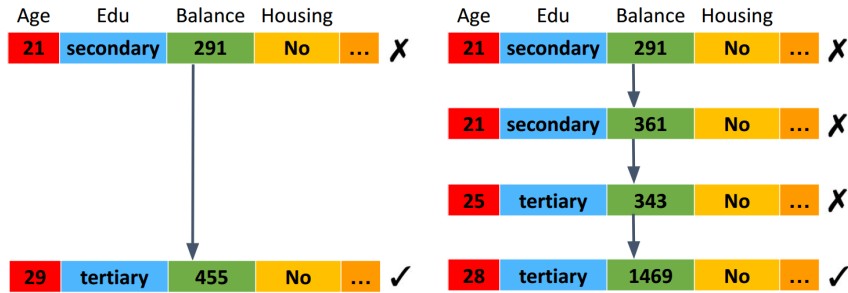

Figure 5: Example of a gradient-based one-step recourse recommendation (left) and a graph-based sequential recourse recommendation (right) on the Bank dataset. ✗ denotes the unfavorable outcomes, and ✓ denotes the favorable outcomes.

## B.2 IMPLEMENTATION DETAILS

**Classifier.** We train a three-layer MLP with 20, 50, and 20 nodes and a ReLU activation function in each layer for each dataset. We report the accuracy and AUC of the underlying classifier for each dataset in Table 4.

Table 4: Accuracy and AUC of the MLP classifiers on eight datasets.

| Dataset | Synthesis | German | Bank | Student | Adult | COMPAS | GMC | HELOC |
|---|---|---|---|---|---|---|---|---|
| Accuracy | 0.98 | 0.72 | 0.89 | 0.93 | 0.85 | 0.83 | 0.94 | 0.74 |
| AUC | 0.99 | 0.62 | 0.68 | 0.97 | 0.9 | 0.82 | 0.84 | 0.81 |

**Settings for Figure 4.** In this experiment, we fix $\varepsilon = 0.01$ and vary the number of questions as an integer $T \in [0, 10]$.

**Settings for Table 1.** In this experiment, we select a total of $T = 5$ questions for our ReAP framework. We choose $\lambda = 1.0$ and $\alpha = 0.01$ for ReAP and Wachter's method. We use the default setting for the proximity weight and the diversity weight of DiCE with values 0.5 and 1.0, respectively.

**Settings for Table 2.** We follow the implementation of CARLA (Pawelczyk et al., 2021) to construct a nearest neighbor graph with $K = 10$. We choose $T = 5$ questions for our ReAP method.

## B.3 ADDITIONAL NUMERICAL RESULTS

### B.3.1 BENCHMARK OF PROPOSED HEURISTICS

**Comparison between two-option questions and multiple-option questions.** Here, we compare two heuristics for choosing the questions: The recourse-pair heuristics in Section 3.2 and multiple-option heuristics in Appendix A.2. We denote the recourse-pair heuristics as ReAP-2 and multiple-option heuristics as ReAP-K. The setting of this experiment is the same as Figure 4.

Figure 6 demonstrates that as $T$ increases, the mean rank of ReAP-K decreases faster than ReAP-2. Because the complexity of both heuristics is $\mathcal{O}(N \log(N))$, these results indicate that the multiple-option heuristic is more efficient in our adaptive preference learning framework.

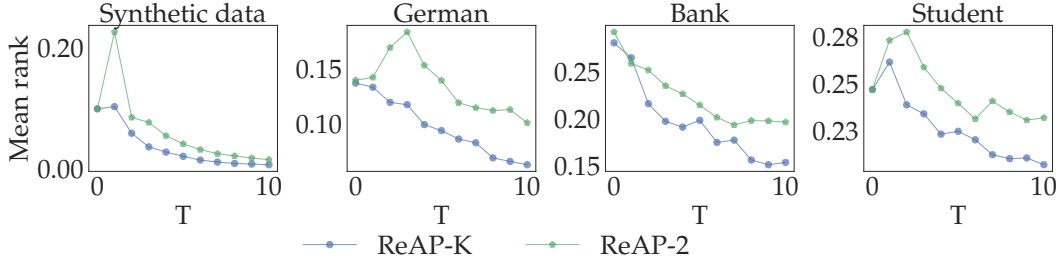

Figure 6: Comparison of two heuristics: recourse-pair heuristics (ReAP-2) and multiple-option heuristics (ReAP-K) with the average mean rank on synthetic data and three real-world datasets.

**Exhaustive search and similar-cost heuristics.** We compare the run time of the similar-cost heuristics and the exhaustive search for a recourse-pair question. This experiment is conducted on a machine with an i7-10510U CPU.

First, we generate $N$ 2-dimensional data samples for each value $N = 100, \ldots, 10000$. Then, for each value of $N$, we compute the average run time of two methods and report the results in Figure 7. We can observe that at $N = 10000$, the exhaustive search requires more than 40s to search for a question, which is impractical in real-world settings.

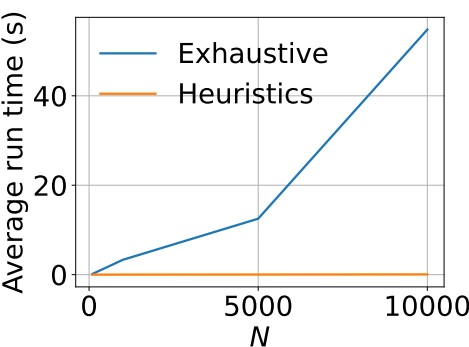

Figure 7: Run time comparison of two heuristics: recourse-pair heuristics (ReAP-2) and multiple-option heuristics (ReAP-K) with the average mean rank on four datasets.

Table 5: The suboptimality gap between the objective of exhaustive search and similar-cost heuristics with inconsistency threshold $\gamma = 0.01$ in four datasets.

|  | Synthetic | German | Bank | Student |
|---|---|---|---|---|
| Gap | 0.001 | 0.017 | 0.021 | 0.029 |

Table 6: Benchmark of Cost and Validity between gradient-based methods on four datasets.

| Dataset | Methods | Cost | Validity |
|---|---|---|---|
| Adult | DiCE | $2.89 \pm 1.42$ | $\mathbf{1.00} \pm 0.00$ |
| | Wachter | $0.06 \pm 0.04$ | $\mathbf{1.00} \pm 0.00$ |
| | ReAP | $\mathbf{0.04} \pm 0.05$ | $\mathbf{1.00} \pm 0.00$ |
| COMPAS | DiCE | $0.51 \pm 1.32$ | $\mathbf{1.00} \pm 0.00$ |
| | Wachter | $\mathbf{0.03} \pm 0.04$ | $\mathbf{1.00} \pm 0.00$ |
| | ReAP | $\mathbf{0.03} \pm 0.04$ | $\mathbf{1.00} \pm 0.00$ |
| GMC | DiCE | $0.25 \pm 0.16$ | $\mathbf{1.00} \pm 0.00$ |
| | Wachter | $0.02 \pm 0.01$ | $\mathbf{1.00} \pm 0.00$ |
| | ReAP | $\mathbf{0.01} \pm 0.00$ | $\mathbf{1.00} \pm 0.00$ |
| HELOC | DiCE | $0.43 \pm 0.22$ | $\mathbf{1.00} \pm 0.00$ |
| | Wachter | $\mathbf{0.05} \pm 0.07$ | $\mathbf{1.00} \pm 0.00$ |
| | ReAP | $\mathbf{0.05} \pm 0.06$ | $\mathbf{1.00} \pm 0.00$ |

Let $\mathrm{obj}_e$ and $\mathrm{obj}_h$ be the optimal values for exhaustive search and similar-cost heuristics, respectively. We compare the relative suboptimality gap between the objective of those two methods as the following:

$$\mathrm{gap}(\mathrm{obj}_e, \mathrm{obj}_h) = \frac{|\mathrm{obj}_e - \mathrm{obj}_h|}{\mathrm{obj}_e}.$$

The experiment results show that the suboptimality gap between the objective of the two methods is approximately of order $10^{-6}$ in all datasets. These results demonstrate empirically that our heuristic method can generate good solutions to the original problem at a fraction of the computational time.

**Heuristics to address human inconsistencies.** To account for similarity-dependent uncertainty, we can adapt our heuristics by taking into consideration only an adjacent pair of $([i], [i+1])$ for $i \in [\![N]\!]$ if the disparity between their costs is larger than an inconsistency threshold, denoted as $\gamma$.

We compare the objective values, in terms of their difference, of those two methods in Table 5. These results demonstrate that the proposed heuristic method can generate reasonable solutions to the original problem at a fraction of the computational time compared to the exhaustive search.

### B.3.2 RESULTS ON MORE DATASETS

Here, we report the additional numerical results for four datasets available in CARLA (Pawelczyk et al., 2021), including Adult, COMPAS, GMC, and HELOC. We report the results in Table 6 and Table 7. These results demonstrate that our method outperforms other baselines, effectively adjusts to the subject's cost function, and adequately reflects the individual subject's preferences.

### B.3.3 COMPARISON WITH PEAR

We implement the PEAR method proposed by De Toni et al. (2023) based on our understanding of the method and the details outlined in the original paper.[1] We conduct this experiment using Adult and GMC datasets, consistent with their usage in De Toni et al. (2023).

Comparing our method to PEAR (De Toni et al., 2023) is not straightforward due to the difference in the cost modeling. Specifically, De Toni et al. (2023) utilizes a linear structural causal model, whereas we assume the cost function takes the form of the Mahalanobis distance. In this experiment, we employ a Manhattan ($\ell_1$) distance to measure the cost of the actions. In this way, both our method and the PEAR method are misspecified. This experiment aims to benchmark which method is more robust to the misspecification of the cost functional form. As we assume that the subject $x_0$ has the true cost function of the form $c(x, x_0) = \|x - x_0\|_1$, between two recourses $x_i$ and $x_j$, $x_i$ is preferred to $x_j$ if

$$\|x_i - x_0\|_1 \leq \|x_j - x_0\|_1.$$

---

[1]As of submission's date, the implementation for De Toni et al. (2023) has not been publicly released.

Table 7: Benchmark of Path cost between graph-based ReAP and FACE. All methods attain the validity of $1.00 \pm 0.00$. Thus, we do not display Validity in the table.

| Dataset | Methods | Path cost |
|---------|---------|-----------|
| Adult | FACE | $0.77 \pm 0.56$ |
| | ReUP | $\mathbf{0.75} \pm 0.52$ |
| COMPAS | FACE | $0.93 \pm 0.75$ |
| | ReUP | $\mathbf{0.79} \pm 0.61$ |
| GMC | FACE | $\mathbf{0.61} \pm 0.49$ |
| | ReUP | $0.65 \pm 0.42$ |
| HELOC | FACE | $1.05 \pm 0.76$ |
| | ReUP | $\mathbf{0.95} \pm 0.65$ |

Table 8: Benchmark of Path cost between PEAR and graph-based ReAP. All methods attain the validity of $1.00 \pm 0.00$. Thus, we do not display Validity in the table.

| Dataset | Methods | Path cost $\downarrow$ |
|---------|---------|------------------------|
| Adult | PEAR | $1.78 \pm 0.91$ |
| | ReAP | $\mathbf{1.76} \pm 1.02$ |
| GMC | PEAR | $0.96 \pm 0.52$ |
| | ReAP | $\mathbf{0.81} \pm 0.39$ |

Our approach employs the above response model for the construction of the terminal confidence set $\mathcal{U}_{\mathbb{P}}$. In contrast, PEAR utilizes the same model for the selection of the optimal intervention in each iteration (De Toni et al., 2023, Algorithm (1)). Regarding the objective, our method is designed to learn the matrix $A_0$ within the framework of Mahalanobis distance while PEAR's objective is to learn the optimal weights for the cost function outlined in De Toni et al. (2023, Equation (3)).

We choose $T = 10$ questions for both methods to ensure a fair comparison. Additionally, since our approach involves pairwise comparisons between recourses, we set the choice set size for PEAR to 2, which aligns with our method. Following the settings in De Toni et al. (2023), the prior distribution of weights takes the form of a mixture of Gaussians with 6 components, where the means were randomized, and the covariance matrix was set to identity. When $T = 0$, the weights are initialized using the expected prior value.

After we have learned the cost function using each method, we use the graph-based recourse method wherein we construct the graph using the methodology outlined in the FACE method (Poyiadzi et al., 2020). FACE initiates by constructing a k-NN graph denoted as $\mathcal{G} = (\mathcal{V}, \mathcal{E})$, which serves as a representation of the underlying data's geometry. The graph's vertices correspond to the sampled instances, specifically the training data, while edges establish connections between instances that are in proximity based on the Euclidean distance metric. This closeness measure is encoded in the weights assigned to the edges. Subsequently, for our method, we proceed to reassign the weights of the edges $(x_i, x_j) \in \mathcal{E}$ within the graph, employing the cost function $\bar{w}_{ij} = c_{A^\star}(x_i, x_j)$, where $A^\star$ is the Chebyshev center of the terminal confidence set. For PEAR, we reassign the edge weights using the cost function defined in De Toni et al. (2023, Equation (3)). Thus, the two methods can access the same graph structure but different edge costs. We then solve the graph-based recourse problem in Section 4.2. Finally, we evaluate the path cost from input $x_0$ to $x_r$, evaluated with Manhattan distance, which is the true cost function in this experiment.

Table 8 reports the mean and standard deviation of path cost over 100 test samples. These results demonstrate that our method ReAP performs comparable to PEAR in the Adult dataset, while we outperform PEAR in the GMC dataset.

## C  PROOFS

We here provide the proof of Theorem 3.1 and Proposition 4.1 that are omitted in the main text.

### C.1  PROOF OF THEOREM 3.1

*Proof of Theorem 3.1.* The proof of Theorem 3.1 follows a similar line of argument as Theorem A.1, in which we consider the form

$$
\begin{aligned}
\max \quad & r \\
\text{s.t.} \quad & A_c \in \mathbb{S}_+^d, \ r \in \mathbb{R}_+ \\
& \langle A_c + \Delta, M_{ij} \rangle \leq \varepsilon \ \forall \Delta \in \mathcal{B}_r, \ \forall (i,j) \in \mathbb{P}.
\end{aligned}
$$

A similar reformulation to Theorem A.1 using the dual norm leads to the postulated result.  □

### C.2  PROOF OF PROPOSITION 4.1

*Proof of Proposition 4.1.* Semidefinite programming duality asserts that

$$
\max_{A \in \mathcal{U}_{\mathbb{P}}} \sum_{(i,j) \in \mathcal{E}} w_{ij}(A) f_{ij} =
\begin{cases}
\max & \left\langle A, \sum_{(i,j) \in \mathcal{E}} (x_i - x_j)(x_i - x_j)^\top f_{ij} \right\rangle \\
\text{s.t.} & \langle A, M_{ij} \rangle \leq \varepsilon \ \forall (i,j) \in \mathbb{P} \\
& 0 \preceq A \preceq I
\end{cases}
$$

$$
=
\begin{cases}
\min & \langle U, I \rangle + \varepsilon \sum_{(i,j) \in \mathbb{P}} t_{ij} \\
\text{s.t.} & U + \sum_{(i,j) \in \mathbb{P}} M_{ij} t_{ij} \succeq \sum_{(x_i, x_j) \in \mathcal{E}} (x_i - x_j)(x_i - x_j)^\top f_{ij} \\
& t_{ij} \geq 0 \ \forall (i,j) \in \mathbb{P}, \ U \in \mathbb{S}_+^d.
\end{cases}
$$

Replacing the minimization above into the objective function leads to the postulated result.  □

## D  MOTIVATION FOR THE MAHALANOBIS COST FUNCTION

We provide two arguments to support the choice of the Mahalanobis cost function. The first argument involves a control theory viewpoint, while the second argument is the connection with the structural causal model.

### D.1  LINEAR QUADRATIC REGULATOR COST

In this section, we describe a sequential control process that affects feature transitions of a subject $x_0$ towards a target feature while minimizing the cost of efforts. Let $x_0$ and $x_r$ be the initial feature of the subject and the target feature. We consider a discrete-time system that, at each iteration, an input effort $u^{(t)}$ drives $x^{(t)}$ to $x^{(t+1)}$

$$
x^{(t+1)} = x^{(t)} + u^{(t)}, \quad x^{(0)} = x_0.
$$

The objective is to finding the best input efforts $u^{(t)}$ ($\forall t = 0, \ldots, \infty$) to move from $x_0$ toward $x_r$. One can formulate this as solving a Linear Quadratic Regulator (LQR) problem of the form:

$$
c(x_0, x_r) =
\begin{cases}
\min & \sum_{t=0}^\infty (x^{(t)} - x_r)^\top Q (x^{(t)} - x_r) + (u^{(t)})^\top R u^{(t)} \\
\text{s.t.} & u^{(t)} \in \mathbb{R}^d \quad \forall t = 0, \ldots, \infty \\
& x^{(t+1)} = x^{(t)} + u^{(t)} \quad \forall t \\
& x^{(0)} = x_0,
\end{cases}
$$

where the parameters $Q$ and $R$ are the subject's state cost and input cost matrices, respectively. The matrix $Q$ is positive semidefinite symmetric while $R$ is positive definite symmetric. The value $c(x_0, x_r)$ is the cost to implement the recourse $x_r$.

**Proposition D.1** (Quadratic cost). *The optimal cost function $c(x_0, x_r)$ is quadratic, that is:*

$$
c(x_0, x_r) = (x_0 - x_r)^\top A_0 (x_0 - x_r),
$$

*where $A_0$ is a positive definite symmetric matrix satisfying the following equation:*

$$
Q = A_0 (R + A_0)^{-1} A_0.
$$

Proposition D.1 asserts that the minimal cost function has the Mahalanobis form, which solely relies on the initial input $x_0$ and the target features $x_r$.

*Proof of Proposition D.1.* Because $x_r$ is a fixed vector, use the following change of variables $z^{(t)} \leftarrow x^{(t)} - x_r$, we have the equivalence

$$c(x_0, x_r) = \begin{cases} \min & \sum_{t=0}^{\infty} (z^{(t)})^\top Q z^{(t)} + (u^{(t)})^\top R u^{(t)} \\ \text{s.t.} & u^{(t)} \in \mathbb{R}^d \quad \forall t = 0, \ldots, \infty \\ & z^{(t+1)} = z^{(t)} + u^{(t)} \quad \forall t \\ & z^{(0)} = x_0 - x_r. \end{cases}$$

Let $V(z)$ be the minimum LQR cost-to-go, starting from state $z$ as follows:

$$V(z) = \begin{cases} \min & \sum_{t=0}^{\infty} (z^{(t)})^\top Q z^{(t)} + (u^{(t)})^\top R u^{(t)} \\ \text{s.t.} & u^{(t)} \in \mathbb{R}^d \quad \forall t = 1, \ldots, \infty \\ & z^{(t+1)} = z^{(t)} + u^{(t)} \quad \forall t \\ & z^{(0)} = z. \end{cases}$$

According to Bertsekas (2012, Section 4.1), the function $V$ has a quadratic form $V(z) = z^\top A_0 z$, for some symmetric matrix $A_0$. Because $Q$ is a positive semidefinite symmetric matrix and $R$ is a positive definite symmetric matrix, $V(z) > 0$ for all $z \in \mathbb{R}^d$, meaning that $A_0$ is a positive definite symmetric matrix. Substituting $\sum_{t=1}^{\infty} (z^{(t)})^\top Q z^{(t)} + (u^{(t)})^\top R u^{(t)}$ by $V(z + u^{(0)})$, we have:

$$V(z) = \min_{u^{(0)}} z^\top Q z + (u^{(0)})^\top R u^{(0)} + V(z + u^{(0)}),$$

which implies that

$$z^\top A_0 z = \min_{u^{(0)}} z^\top Q z + (u^{(0)})^\top R u^{(0)} + (z + u^{(0)})^\top A_0 (z + u^{(0)}).$$

It is easy to see that the objective function of the right-hand side optimization problem is convex. Therefore, the optimal solution of $u^{(0)}$ satisfies

$$R u^{(0)} + A_0 (z + u^{(0)}) = 0 \implies u^{(0)*} = -(R + A_0)^{-1} A_0 z.$$

Here the inversion of $(R + A_0)$ is feasible because $R$ and $A_0$ are positive definite matrices. Then we have:

$$z^\top A_0 z = z^\top Q z + (u^{(0)*})^\top R u^{(0)*} + (z + u^{(0)*})^\top A_0 (z + u^{(0)*})$$
$$\Leftrightarrow z^\top A_0 z = z^\top (Q + A_0 - A_0 (R + A_0)^{-1} A_0) z.$$

Therefore, the matrix $A_0$ needs to satisfy the following condition:

$$A_0 = Q + A_0 - A_0 (R + A_0)^{-1} A_0 \Leftrightarrow Q = A_0 (R + A_0)^{-1} A_0.$$

This completes our proof. □

**Remark D.2** (Finite time horizon). *The argument in this section relies on an infinite horizon control problem to simplify the discussion. One can formulate a similar finite horizon problem, which leads to a similar Mahalanobis form. The proof follows from an induction argument, which is standard in the control theory literature; see Bertsekas (2012).*

### D.2 CASUAL GRAPH RECOVERY

This section discusses the connection between the linear Gaussian structural causal model and the Mahalanobis cost function. We consider a linear Gaussian structural equation model ($SEM$) for the deviation $\delta \in \mathbb{R}^d$ from the initial input $x_0 \in \mathbb{R}^d$ as follows:

$$\delta \sim SEM(W_0, D_0) \Leftrightarrow \delta = W_0 \delta + \epsilon, \tag{11}$$

where $\epsilon \sim \mathcal{N}(0, D_0)$ is a multivariate Gaussian with mean vector zero and a covariance matrix $D_0 \in \mathbb{S}_{++}^d$. The $W_0 \in \mathbb{R}^{d \times d}$ is equivalent to the weight $w$ of the structural causal model (SCM) for cost formulation from a directed acyclic graph (DAG) $G$. Each node of $G$ is associated with a single

feature, and a nonzero entry $(W_0)_{i,j}$ corresponds to a causal relationship from node $j$ to node $i$. The SEM (11) implies that:

$$\delta \sim \mathcal{N}(0, (I - W_0)^{-1} D_0 (I - W_0)^{-\top}),$$

where $I$ is the identity matrix. The density function for $\delta$ is:

$$f_0(\delta) = \frac{1}{(2\pi)^{d/2} |(I - W_0)^{-1} D_0 (I - W_0)^{-\top}|^{1/2}} \exp\left(-\frac{1}{2} \delta^\top (I - W_0)^\top D_0^{-1} (I - W_0) \delta\right).$$

Between two deviations $\delta_i = x_i - x_0$ and $\delta_j = x_j - x_0$, the subject prefers a deviation with a higher likelihood, and thus $\delta_i$ is preferred to $\delta_j$ if

$$\delta_i^\top (I - W_0)^\top D_0^{-1} (I - W_0) \delta_i \leq \delta_j^\top (I - W_0)^\top D_0^{-1} (I - W_0) \delta_j.$$

We recover the Mahalanobis cost preference model with $A_0$ corresponding to the precision matrix of the deviation under the linear Gaussian structural equation model. Specifically, the value of $A_0$ is computed as

$$A_0 = (I - W_0)^\top D_0^{-1} (I - W_0).$$

