# OpenReview forum: "Cost Adaptive Recourse Recommendation by Adaptive Preference Elicitation"
_ICLR.cc/2024/Conference — Submitted to ICLR 2024_

### Official Review · Reviewer_jnH1 · 2023-10-29

**Soundness:** 3 good
**Presentation:** 4 excellent
**Contribution:** 3 good
**Rating:** 6
**Confidence:** 3

**Summary:**

The work addresses the problem of personalized algorithmic recourse
where the cost function of the user has to be learned by interacting
with the user. The problem is formalized as learning a Mahalanobis
distance in state space, and combining it with two existing approaches
for differentiable and non-differentiable classifiers respectively.

**Strengths:**

In the cases in which the use is assumed to provide consistent
feedback, the approach is technically sound, and it is nicely adapted
to deal with both differentiable and non-differentible classifiers.

The manuscript is well written and clear, and the algorithmic solutions are well motivated.

The authors made a substantial effort in comparing with existing
alternatives, including a recent approach (De Toni et al, 2022) for
which no implementation is currently available.

**Weaknesses:**

User inconsistencies are dealt in a less principled way in my
opinion. The authors do provide in the supplement a (computationally
challenging) solution to account for inconsistencies. However this
assumes to have a given max percentage of inconsistencies. Approaches
to account for user inconsistencies typically assume a user response
model, where the probability of a wrong preference feedback grows with
the similarity between instances. This is one of the advantages of a
Bayesian approach to preference elicitation.

The approach provides cost-learning functionalities on top of two
existing approaches, a differentiable and a non-differentiable one. It
thus inherits the limitations of these approaches, e.g. for the
non-differentiable approach, the fact that recourse can only be
achieved by moving to an example in the training set that achieves
recourse. This is suboptimal, as a new user could achieve recourse in
a way that is different from training users and has lower cost. These
limitations should also be mentioned.

**Questions:**

Is it possible to incorporate a user response model modelling uncertainty in feedback?

Did you study how your approach behaves when faced with inconsistent users? This is major advantage of Bayesian approaches to preference elicitation..

---

> ### Author Response · Authors · 2023-11-21
> **Response to Reviewer jnH1**
>
> Dear reviewer,
>
> We thank you for the review and thoughtful feedback!
>
> From our understanding, the reviewer is proposing that we should employ a Bayesian approach. We agree that the Bayesian approach is a flexible and systematic framework to deal with this cost-learning problem. In fact, the reference PEAR (De Toni et al. (2022)) relies on a Bayesian framework to learn the cost function. However, the Bayesian approach is computationally intensive, and many ad-hoc adaptations need to be used in order to make it work (sampling, variational posterior, etc.). All components of the Bayesian method, including computing the posterior distribution and identifying the questions to ask the subject, require integration, which is computationally expensive.
>
> Our ReAP method focuses on tractability. That is why we focus on a deterministic approach to solving the cost learning problem: in a noiseless setting, finding the center is done by semidefinite programming, and finding the questions is done by linear algebra computations. All components in our paper are computationally tractable. Hence, at least, we have a solution method that can be deployed in an efficient manner. Empirically, we also show that our method performs well in popular datasets.
>
> Considering uncertainty in user response is very important, we totally understand that. In a deterministic setting, we have shown how to hedge against this uncertainty using binary variables. We report the numerical results in the table below. The table demonstrates that our method outperforms FACE in terms of cost, indicating that our approach efficiently captures the preferences considering uncertainty in user response.
>
> | Datasets  |Methods               |   Path cost               |
> |----------|-----------------------------------------|---------------------|
> |German | FACE |  0.66 $\pm$ 0.48  |
> |    | ReAP  |  0.51 $\pm$ 0.46  |
> |Bank | FACE |  1.20 $\pm$ 0.69  |
> |    | ReAP |  0.95 $\pm$ 0.57  |
> |Student | FACE |  1.10 $\pm$ 0.76  |
> |   | ReAP |  0.93 $\pm$ 0.62  |
>
> Developing a Bayesian method is an attractive idea. However, we believe that it is too far away from the narrative and the goal of this paper. A Bayesian approach will significantly differ from everything we present in this paper. Thank you!

---

### Official Review · Reviewer_Qous · 2023-10-30

**Soundness:** 3 good
**Presentation:** 3 good
**Contribution:** 2 fair
**Rating:** 5
**Confidence:** 4

**Summary:**

The paper presents ReAP, a method to generate personalised recourse suggestions via preference elicitation. The authors provide a technique to learn a personalized cost function, for each user, by asking preference questions. The authors also show extended versions of a gradient-based and graph-based approach for recourse that exploit their learned cost function. Lastly, they validate ReAP with some experiments.

**Strengths:**

The topic of the paper is timely, and I think it is a direction which needs to be explored by the recourse community in order to make this field progress (and more realistic). The paper is also clear in its exposition, the idea is well-motivated and the formalization of the problem is straightforward.

**Weaknesses:**

The main concern is that I do not see the advantages of this approach compared to the closets-related method PEAR [1]. While it is true that they seem to rely on some sort of causal graph, which might be hard to get, their technique can be in principle applied to different cost functions (even the one presented here) and they provide a Bayesian model to incorporate uncertainty over the user answers. Therefore, it is not clear to me the benefits of ReAP over PEAR.

In the introduction, the author says “ [...] our framework can perform well even when the dimension of the feature space grows large [...]”. First, this assertion is true only under a synthetic setting. Real-world users will find it difficult to compare profiles with too many features. For this reason, I would have expected to see experiments on “noisy” users, which is a common scenario in the preference elicitation literature [2]. Equation 2 seems to accommodate it, but I think it would have made the evaluation less synthetic to see some results concerning the role of $\epsilon$.

Lastly, ReAP disentangle the elicitation part from the recourse generation (Figure 1). However, if the number of features is truly high, we might just need to estimate only the cost of the features we need for recourse, rather than the full matrix $A$. Moreover, it might be desirable that we just ask the user the right questions to estimate the perfect recourse for him/her, rather than the ones minimizing the smallest projection distance.

[1] De Toni, Giovanni, et al. “Personalized Algorithmic Recourse with Preference Elicitation" arXiv preprint arXiv:2205.13743 (2022). (it appears a new version of the paper was published the last May).
[2] Viappiani & Boutilier. "Optimal bayesian recommendation sets and myopically optimal choice query sets." NeurIPS (2010).

**Questions:**

* Why did the author not include PEAR as a baseline in the main paper (while it is present in the Appendix)? I think it is the most suitable competitor since its method can be applied in principle to all cost functions, it employs a more targeted approach to the recourse pair selection and it deals with the user’s uncertainty in a most principled way.

* How do you explain the fact that Watcher and ReAP achieve almost the same cost/validity in all four datasets (Table 1)?

* In Figure 4, would it be possible to show the results also for a random strategy to select the recourse pairs to show the user? It is a common check to understand if the experimental setting is not too simple (considering also the concerns of Table 1).

* What happens if the users are “noisy” in their answers? For example, they gave a wrong answer to the preference questions. Equation 2 seems to accommodate this issue, but I do not see any experiments showing the effect of $\epsilon$ in the cost estimation quality and/or final recourse.

* What is the novelty of the approaches presented in Sections 4.1 and 4.2? Especially section 4.1 seems a trivial extension of Watcher et al.

---

> ### Author Response · Authors · 2023-11-21
> **Response to Reviewer Qous [1/2]**
>
> Dear reviewer,
>
> We appreciate that you review our paper and provide insightful comments. We would like to address your concerns as follows:
>
> > Therefore, it is not clear to me the benefits of ReAP over PEAR.
>
> There are several benefits of ReAP:
> 1. ReAP is computationally tractable. ReAP is based on semidefinite programming and geometric calculations to find the questions. All operations in ReAP can be done in less than half a second on a laptop. In contrast, the Bayesian approach proposed by De Toni et al. (2022) is computationally intensive.
> 2. ReAP delivers a better solution than PEAR. In Table 8, we have shown that our method outperforms PEAR in the Adult and GMC datasets.
> 3. It is not trivial to generalize PEAR to the Mahalanobis cost setting while using the Mahalanobis distance as in our paper can bridge with causal models, as we described in Appendix D.2.
> 4. PEAR requires a linear structural causal model to define the cost of changing a specific feature, whereas our method does not require such a model.
>
> > Why did the author not include PEAR as a baseline in the main paper
>
> Thank you for your comment. We include PEAR in the appendix for a better explanation of the formulation and the settings of the experiments. We will include these results in the main paper in our final version, in which more space is allowed.
>
>
> > If the number of features is truly high, we might just need to estimate only the cost of the features we need for recourse rather than the full matrix
>
> We agree that we only need to estimate the cost associated with the features we need for recourse. However, the number of features we need for recourse can also be truly high. In this case, the number of parameters to be estimated/learned is still large.
>
> > How do you explain the fact that Watcher and ReAP achieve almost the same cost/validity in all four datasets?
>
> Regarding validity, we employ an early stopping criterion, terminating the algorithm when a valid recourse is obtained for both algorithms. Therefore, both methods have a perfect (1.0) validity.
>
> The objective function for Wachter is non-convex for a fixed $\lambda$ when we consider an MLP classifier. In particular, if we initialize $\lambda=0$, and then increase the value of $\lambda$, the difference in the cost of recourse compared to the initial $\lambda$ is negligible. Therefore, Watcher and ReAP achieve almost the same cost. However, our methods still show improvements over Wachter in terms of cost in all the experiments.

---

> > ### Author Response · Authors · 2023-11-21
> > **Response to Reviewer Qous [2/2]**
> >
> > > In Figure 4, would it be possible to show the results also for a random strategy to select the recourse pairs to show the user?
> >
> > Here, we show the comparison of mean rank between the random strategy to select the recourse pairs and the proposed heuristics for $T=0$, $T=5$, and $T=10$:
> >
> > | Datasets  |Methods               |   $T=0$            |$T=5$               |$T=10$               |
> > |----------|-----------------------------------------|---------------------|---------------------|---------------------|
> > |German | Random |  0.26  | 0.24    | 0.15    |
> > |    | Ours |  0.13  | 0.10    | 0.07    |
> > |Bank | Random |  0.42  | 0.31    | 0.19    |
> > |    | Ours |  0.27  | 0.20    | 0.15    |
> > |Student | Random |  0.37  | 0.31    | 0.29    |
> > |   | Ours |  0.25  | 0.23    | 0.21    |
> >
> > These results show that as $T$ increases, the mean rank of our approach decreases faster than the random strategy, indicating that the proposed heuristic is more efficient in our adaptive preference learning framework.
> >
> > > Experiments showing the effect of $\epsilon$ in the cost estimation quality and/or final recourse.
> >
> > We show the effect of $\epsilon$ on the cost final recourse  for the German dataset in the following table:
> >
> > | Datasets  |Methods               |   Path cost         |
> > |----------|-----------------------------------------|---------------------|
> > |German | FACE |  0.66 $\pm$ 0.48  |
> > | | ReAP ($\epsilon=0.01$) |  0.53 $\pm$ 0.49  |
> > |    | ReAP ($\epsilon=0.02$) |  0.55 $\pm$ 0.51  |
> > |    | ReAP ($\epsilon=0.05$) |  0.51 $\pm$ 0.45  |
> > |    | ReAP ($\epsilon=0.1$) |  0.51 $\pm$ 0.45  |
> >
> > These results indicate that the path cost shows only minor variations across different values of $\epsilon$. Moreover, our method continues to outperform FACE.
> >
> > > What is the novelty of the approaches presented in Sections 4.1 and 4.2?
> >
> > In Section 4.1 and 4.2, we generalize the two existing counterfactual generation methods: Wachter and FACE. In Section 4.1, We find a recourse that balances the trade-off between the validity and the worst-case cost. In Section 4.2, we aim to minimize the total cost of the sequential recourse subject to the most unfavorable scenario of $A_0$ within the final confidence set. Optimizing over the worst-case objective is novel and has several advantages:
> >
> > **Optimize over worst-case objective.** The terminal set $\mathcal U_{\mathbb{P}}$ is generally not a singleton as it still contains many possible matrices that conform with the feedback information. To find the recourse, we need to borrow the ideas from robust optimization, which formulates the problem as a min-max optimization problem. Looking at the worst-case situation can eliminate any bad surprises regarding the implementation cost. Practically, the min-max problem manages to provide an acceptable recourse for challenging scenarios within the domain where $A$ is a matrix satisfying $A \preceq I, A \in \mathbb{S}_{+}^d$. This approach also proves valuable when users' responses contain significant noise and inconsistencies, which in turn results in a still large search space for $A_0$ in the final round.

---

### Official Review · Reviewer_r13q · 2023-10-30

**Soundness:** 4 excellent
**Presentation:** 4 excellent
**Contribution:** 2 fair
**Rating:** 8
**Confidence:** 4

**Summary:**

This paper proposes a method for learning a human subject’s recourse preferences, so as to deliver better options in the problem of algorithmic recourse. Then, methods are presenting for providing recourses given a learned cost function. The paper is technically innovative, and the approach it presents is very reasonable. The method assumes a Mahalanobis cost function for the user, and it gradually restricts the space of cost functions that is consistent with queried user preferences. Query selection is done to find aggressive cuts of the candidate set space in order to efficiently reduce the candidate set size. Existing gradient-based and blackbox (graph-based) recourse selection algorithms are adapted to using the learned cost functions. Experiments are conducted to compare the proposed method with two common baselines, Wachter et al 2018 and DiCE. It’s shown that the method does reduce ground-truth user cost over time (using simulated, identifiable cost functions) and the method does at least as well as baselines.

**Strengths:**

- Very Important: The proposed method tackles an important problem in a reasonable way. It’s clear that algorithmic recourse mechanisms should be able to leverage subject feedback/preferences to improve recourse quality. The proposed method makes reasonable choices at every turn as it tackles this problem. The cost function family is well motivated, and it is nice that this method reduces to simpler Euclidean-distance-based approaches at T=0 queries. The cost inference pipeline is naturally integrated with different optimization procedures for recourse selection (although this could take some work, as evidenced by the integration with FACE). The approximation to the O(n^2) search in cost inference is not only reasonable but actually a nice feature of the method, since it encodes a bias for selecting pairs where there could be a lot of information gain in the query.
- Important: The development of sequential recourses when combining the method with FACE is technically impressive and likely to be useful for downstream applications with real subjects.
- Important: The evaluation shows that the proposed method does learn user preferences over time and thereby lowers ground-truth user cost over additional queries with simulated ground truth preferences. On a number of datasets, the method works at least as well as past baselines.
- Important: The paper provides a lot of context to its approach through extensive connections to related work.

**Weaknesses:**

- Important: So, why wasn’t a comparison conducted against Rawal and Lakkararaju (2022)? I agree that the Bradley-Terry cost model is not expressive enough to capture many realistic cost functions, but then again this criticism should apply to the Mahalanobis family as well. I do not understand why it is claimed without further substantiation that the proposed method would perform better in high dimensions than the Bradley-Terry model, and so I worry that this is a key missing baseline in the current work. It could be that their method works as well as as the ReAP, while being conceptually simpler than ReAP.
- Important: Another question about the method is its limited improvements over even simple baselines on common datasets. There is often no clear improvement over a method from 2018. I wonder if this is an artifact of the simple ground-truth cost function distribution — could this distribution produce more heterogenous cost functions that better separate methods (particularly by being poorly approximated by a simple Euclidean distance)? Regardless, this is a mark against the method. While there are more noticeable improvements against FACE on the sequential recourses (Table 2), it’s not clear whether these are statistically significant.

**Questions:**

- What is the size of the solution set returned by DiCE? Is the comparison with ReAP fair in terms of computational cost and subject query cost?
- Comment: You might also reference https://arxiv.org/pdf/2111.01235.pdf, which investigates using a distribution over plausible cost functions to find a set of recourses that could better satisfy a user.
- Why optimize over the worst case cost function? How might results vary based on the ground truth cost function distribution and the choice of optimizing over the worst case cost function vs some centroid cost function or expected cost function.

---

> ### Author Response · Authors · 2023-11-21
> **Response to Reviewer r13q**
>
> Dear reviewer,
>
> We thank you for reviewing and providing constructive feedback. Below, we address your concerns.
>
> > Why wasn’t a comparison conducted against Rawal and Lakkararaju (2022)?
>
> Rawal and Lakkararaju (2022) employ the Bradley-Terry model to estimate a universal cost function (one cost function for all users) and then utilize the user input to generate personalized recourse for the user. The setting of Rawal and Lakkararaju (2022) is significantly different from ours. Our setting requires learning a personalized cost function through a preference elicitation process (each user is associated with a different, personalized cost function). Therefore, the comparison between the two methods is not necessarily fair.
>
> Moreover, the method from Rawal and Lakkaraju (2020) is addictive in features. Therefore, it is unable to recover the underlying causal graph. As in our paper, using the Mahalanobis distance can bridge with causal models, as described in Appendix D.2.
>
> > The method is its limited improvements over even simple baselines on common datasets
>
> We do not agree with this observation. In Table 1 and Table 2, we demonstrate that our method significantly outperforms DiCE and FACE in terms of cost. Our method even outperforms the adaptive version of Wachter on three over four datasets.
>
> > What is the size of the solution set returned by DiCE? Is the comparison with ReAP fair in terms of computational cost and subject query cost?
>
> In our experiments, we find a \textbf{single recourse} for each method to ensure a fair comparison. We report the computational time comparison between our method and DiCE in the table below.  Our method has a significantly lower runtime compared to DiCE.
>
> | Dataset  |German               |Bank               |Student               |
> |----------|-----------------------------------------|---------------------|---------------------|
> | DiCE | 0.87 |  0.89  | 1.19    |
> | ReAP | 0.23 | 0.25  | 0.31    |
>
>
>
> >  Reference
>
> Thank you for your comment. We have included the recommended reference in our revised draft.
>
> > Why optimize over the worst-case cost function?
>
> The terminal set $\mathcal U_{\mathbb{P}}$ is generally not a singleton as it still contains many possible matrices that conform with the feedback information. To find the recourse, we need to borrow the ideas from robust optimization, which formulates the problem as a min-max optimization problem. Looking at the worst-case situation can eliminate any bad surprises regarding the implementation cost. Practically, the min-max problem manages to provide an acceptable recourse for challenging scenarios within the domain where $A$ is a matrix satisfying $A \preceq I, A \in \mathbb{S}_{+}^d$. This approach also proves valuable when users' responses contain significant noise and inconsistencies, which in turn results in a still large search space for $A_0$ in the final round.

---

> > ### Comment · Reviewer_r13q · 2023-11-22
> > **Reply to authors**
> >
> > Thanks for the reply. Comments below.
> >
> > > Rawal and Lakkararaju (2022) employ the Bradley-Terry model to estimate a universal cost function (one cost function for all users) and then utilize the user input to generate personalized recourse for the user. The setting of Rawal and Lakkararaju (2022) is significantly different from ours…Moreover, the method from Rawal and Lakkaraju (2020) is addictive in features. Therefore, it is unable to recover the underlying causal graph.
> >
> > Oh sorry thanks, I forgot about the overall population vs. individuals difference. I would say the question remains, if this is a simple/popular preference model, why not compare with Bradley-Terry models fit to individual users as T=1…10 queries are collected? But I see that the Mahalanobis family is ultimately more expressive than the Bradley-Terry model, and that is a clear advantage.
> >
> > > In Table 1 and Table 2, we demonstrate that our method significantly outperforms DiCE and FACE in terms of cost.
> >
> > Well, we might have to disagree here. I’m looking at results like 0.12 ± 0.14 vs 0.10 ± 0.15 and am not left with a lot of confidence that the 0.10 is better. Normally in a noisy setting like this, I’d expect some statistical test. The same goes for Table 2. Is 0.53 ± 0.49 better than 0.66 ± 0.48? The trends lean in the right direction — that’s why I said it seems like the method performs at least as well as baselines — but I ultimately expect some clearer evidence to be presented. I also ancticipate this shouldn’t be too hard to demonstrate! I recommend using some more diverse/interesting synthetic cost functions to show any superiority over past methods (plus more data, when possible).
> >
> > > In our experiments, we find a \textbf{single recourse} for each method to ensure a fair comparison.
> >
> > Right, but normally DiCE works by optimizing a set. I don’t even remember if there are objectives that operate at the individual vector level besides the distance and sparsity objectives in DiCE. So is optimization done with just a single vector too? The reason I bring this up is because, if you’re doing T=10 queries, I think it would be fair to run DiCE with k=10 recourses optimized in parallel.
> >
> > > Looking at the worst-case situation can eliminate any bad surprises regarding the implementation cost. Practically, the min-max problem manages to provide an acceptable recourse for challenging scenarios within the domain
> >
> > Fair point! I’d be interested in seeing an ablation in the appendix comparing worst-case to centroid-cost for optimization purposes.
> >
> > ---
> >
> > Based on the above discussion, I’m keeping my overall score at 6 and raising my confidence from 3 to 4.

---

> ### Author Response · Authors · 2023-11-22
> **Thank you and response to the follow-up comments**
>
> Dear reviewer,
>
> Thank you for being responsive in discussing our paper. We address your remaining concerns as the following:
>
> > Why not compare with Bradley-Terry models fit to individual users as T=1…10 queries are collected?
>
> We are eager to conduct an experiment comparing the two methods. However, we cannot find the public implementation of Rawal and Lakkararaju (2022) on ArXiv, NeurIPS, or the author's website.
>
> > It seems like the method performs at least as well as baselines — but I ultimately expect some clearer evidence to be presented.
>
> Looking at the mean and spread is not an informative way to reach a conclusion.
>
> We propose to look at the paired difference of the cost: for each subject, we compute the ReAP cost and the competing method’s cost. We propose to test the hypotheses:
>
> Null hypothesis: ReAP cost equals the competing method’s cost
> Alternative hypothesis: ReAP cost is *smaller* than the competing method’s cost.
>
> In order to test the above hypothesis, we use a one-sided Wilcoxon signed-rank test to compare the paired cost values. The p-value of the test between ReAP and the baselines is reported in the following table.
>
> | Datasets  |German               |   Bank             | Student |
> |----------|-----------------------------------------|---------------------|---------------------|
> |ReAP-DiCE |  $7e-12$ | $3e-18$  | $0.046$ |
> |ReAP-Wachter |  $0.0029$ | $0.008$ | $0.224$ |
> |ReAP-FACE |  $0.009$  | $5e-8$ | $0.019$ |
>
> | Datasets  |Adult | GMC               |
> |----------|-----------------|---------------------|
> |ReAP-PEAR | $0.422$ | $0.0015$ |
>
> Suppose we choose the significant level at 0.05. The table indicates that ReAP significantly outperforms DiCE and FACE across all datasets. ReAP outperforms Wachter in two out of three datasets, except for the Student dataset [Note that this does not imply that Wachter cost is lower than ReAP cost in the Student dataset]. Additionally, ReAP demonstrates superiority over PEAR in the GMC dataset.
>
>
> > There are objectives that operate at the individual vector level besides the distance and sparsity objectives in DiCE. If you’re doing T=10 queries, I think it would be fair to run DiCE with k=10 recourses optimized in parallel.
>
> Each component in the objective of DiCE: validity and proximity, can operate on a single vector, except for the diversity term. In the implementation of DiCE, when choosing $K=1$ recourse, DiCE simply removes the diversity term. The time comparison between the two methods shows that even when generating $K=1$ recourse, our method still exhibits significantly better runtime than DiCE.
>
>
> > I’d be interested in seeing an ablation in the appendix comparing worst-case to centroid-cost for optimization purposes.
>
> We conducted an experiment computing the average path lengths for the two methods, employing A* or solving the worst-case objective in Problem (7) are presented in the following table. The results indicate that the path lengths obtained by solving the worst-case objective are only marginally higher than the alternative, highlighting the effectiveness of our proposed method.
>
>
> | Dataset  |German               |Bank               |Student               |
> |----------|-----------------------------------------|---------------------|---------------------|
> | ReAP-A* | $3.96$ |  $5.89$   | $5.85$    |
> | ReAP-WC | $4.14$ | $6.07$  |  $5.91$  |

---

> ### Comment · Reviewer_r13q · 2023-11-22
>
> Thanks for the quick response. Based on the statistical analysis and new ablations, I will raise my score from 6 to 8.

---

> > ### Author Response · Authors · 2023-11-22
> > **Thank you**
> >
> > Thank you for your prompt response and for raising the score. We are happy to discuss if you have any further questions.

---

### Official Review · Reviewer_WLdg · 2023-11-01

**Soundness:** 2 fair
**Presentation:** 3 good
**Contribution:** 2 fair
**Rating:** 5
**Confidence:** 3

**Summary:**

This paper proposes an approach to recourse generation when the user’s cost is unknown. To learn the user’s cost, pairwise comparisons between points are used and a space of consistent cost matrices is maintained. A min-max objective is used for choosing a recourse either via a gradient-based or a graph-based approach.

**Strengths:**

* Adding preference elicitation for learning user cost makes a nice extension to previous work
* The paper is clearly written
* The experimental results look encouraging

**Weaknesses:**

* It is hard to understand the effect of some approximations (for example, decomposed maximization over edges). An empirical evaluation can help shed some light on that.
* Some details are missing from the experiments (dimension d etc)

**Questions:**

* Queries are chosen from the positive set D_1 only. Does it make sense to consider points from D_0 or new points that have no label (since this is only for the purpose of preference learning), especially in the case of single recourse?
* The worst-case objective might be too conservative, resulting in longer paths. It would be interesting to compare it at least empirically to other choices (A* for example, or a random A \in U_P).
* Sequential recourse: the problem in Eq (7) doesn’t seem like the real problem that needs to be solved. The set U_P updates after every step, and this changes the argmax A and therefore w_ij. Am I missing anything?
* Computation of \bar{w}_ij: maximizing independently over each edge ij is even more conservative. Is it possible to estimate the difference in computed weights between independent and joint optimization (for some small problems)?
* Section 5.1: what is the dimension d in the experiments? Can you comment on the cost of solving (5) using MOSEK? How does it scale up with d?
* Section 5.3: how is lambda chosen for the gradient-based objective?
* In Table 2, what is the average path length?
* Section 6: “We … extend the heuristics to choose the questions from pairwise comparison to multiple-option questions.” I didn’t see where this was done. Can you point to the right section?

Other questions / comments:
* “addictive” => “additive”
* Typo: Algorithm 1, should be “\lambda > 0” in “parameters”?
* In section 3.2, consider adding an algorithm block instead of enumerating the steps.
* Also, in section 4.2 it would be good to have an algorithm block for the graph-based approach.
* Mean rank is referred to in “Recourse generation” (section 5.1), but defined only later, in 5.2.

**Details Of Ethics Concerns:**

Check overlap with submission 4706.
I think it's actually fine, the papers study the same problem but propose different solutions.

---

> ### Author Response · Authors · 2023-11-21
> **Response to Reviewer WLdg [1/2]**
>
> Dear reviewer,
>
> We appreciate that you review our paper and provide insightful comments. We would like to address your concerns as follows:
>
> > Queries are chosen from the positive set $D_1$ only. Does it make sense to consider points from $D_0$ or new points that have no label?
>
> We select recourse pairs from the positive set D_1 to learn the subject's cost function by comparing two **recourses**. It is also possible to consider points from $D_0$ or new points with no label with minimal change to our learning workflow. However, for the recourse application, asking the subject to compare points in $D_0$ may give the user a false sense of psychological expectation. As such, we only consider the positive set $D_1$ and keep the question as "Between two recourses, which one do you prefer?" so that the users know that the two points under comparison both lead to positive predicted outcomes. This setting will help reduce psychological/cognitive bias to the subject's expectations.
>
> > The worst-case objective might be too conservative, resulting in longer paths. It would be interesting to compare it at least empirically to other choices.
> > In Table 2, what is the average path length?
>
> The average path lengths for the two methods, employing $A^\star$ or solving the worst-case objective in Problem (7) are presented in the following table. The results indicate that the path lengths obtained by solving the worst-case objective are only marginally higher than the alternative, highlighting the effectiveness of our proposed method.
>
>
> | Dataset  |German               |Bank               |Student               |
> |----------|-----------------------------------------|---------------------|---------------------|
> | ReAP-$A^\star$ | 3.96 |  5.89   | 5.85    |
> | ReAP-WC | 4.14 | 6.07  |  5.91  |
>
> >  The problem in Eq (7) does not seem like the real problem that needs to be solved. The set $\mathcal U_{\mathbb{P}}$ updates after every step.
>
> We think that the reviewer may have misunderstood the setting here:
> In deployment, the system will pick a fixed value of $T$, and problem (7) will be solved with $\mathcal U_{\mathbb{P}}$ being the terminal confidence set, obtained after we have asked the subject to compare $T$ rounds. Thus, we solve Eq (7) only once.
>
> In the experiments, we conduct the experiments with varying $T$ to study the effects of the number of comparison rounds on the quality of the recourse recommendation. We do not solve (7) multiple times in our workflow.

---

> > ### Author Response · Authors · 2023-11-21
> > **Response to Reviewer WLdg [2/2]**
> >
> > > Computation of $\bar{w}_{ij}$: maximizing independently over each edge ($i, j)$ is even more conservative. Is it possible to estimate the difference in computed weights between independent and joint optimization?
> >
> > To solve the problem (7) for a small graph, we enumerate all possible flows of set $\mathcal F$, then we solve the inner maximization problem. The optimal solution is the sequential recourse with the lowest worst-case cost. This is a brute-force, exhaustive search method to solve (7). We can compute the difference between the optimal solutions of solving the problem (7) and our proposed relaxation using $\bar{w}_{ij}$.
> >
> > The Jaccard distance is popular to measure the dissimilarity between two sets. The results reported below show that the optimal solutions do not differ significantly between the exhaustive search and the relaxed $\bar{w}_{ij}$ problem.
> >
> > | Dataset  |German               |Bank               |Student               |
> > |----------|-----------------------------------------|---------------------|---------------------|
> > | Jaccard | 0.03 $\pm$ 0.02 | 0.05 $\pm$ 0.06  |  0.05 $\pm$ 0.04  |
> >
> > > What is the dimension d in the experiments? Can you comment on the cost of solving (5) using MOSEK? How does it scale up with d?
> >
> > All experiments are conducted using standard datasets in the recourse literature. We report the dimension $d$ and the average running time of solving (5) in the following table:
> >
> > | Dataset  |German               |Bank               |Student               |
> > |----------|-----------------------------------------|---------------------|---------------------|
> > | Dimension | 5 |  8  | 9    |
> > | Runtime | 0.05| 0.06  | 0.06    |
> >
> > The run time of our method is practical for the preference elicitation applications.
> >
> > > How is lambda chosen for the gradient-based objective?
> >
> > We follow the Wachter implementation of CARLA. First, we initialize $\lambda=1.0$, and then we employ an adaptive scheme for $\lambda$ if no valid recourse is found. In particular, if no recourse is found, we reduce the value of $\lambda$ by 0.05, which is similar to CARLA.
> >
> > > Section 6: “We … extend the heuristics to choose the questions from pairwise comparison to multiple-option questions.” I didn’t see where this was done. Can you point to the right section?
> >
> > We include the multiple-option questions heuristics in Appendix A.2.
> >
> > > Other comments
> >
> > Thank you for your comment. We have included all the changes in our revised version.

---

### Meta-Review · Area_Chair_ds1S · 2023-12-06

**Metareview:**

The paper proposes a method for learning personalized cost functions based on (active) queries of pairwise comparisons, and then do recourse recommendation. The proposed framework is evaluated empirically. The method is based on reducing the space of consistent cost matrices.

Pros. Reviewers like the problem of learning personalized cost functions. The study was well-executed. The authors are quite responsive during the rebuttal, which helped clarify some questions and concerns.

Cons. Concerns are raised about insufficient comparisons with previous work, which leads to questions about effectiveness of the proposed methods.

Eventually, while one reviewer was quite positive about the paper, the remaining reviewers remained mildly positive or mildly negative. The paper remained on the fence.

Concerns on ethical issues. Multiple reviewers mentioned that this paper shares many common features with Submission4706, including the problem studied, the references, and the dataset. In particular, the following reg flags were raised.

1. The two submissions used different wordings of the same problem: PREFACTUAL RECOMMENDATION vs RECOURSE RECOMMENDATION.

2. The introductions, especially the motivation, of the two papers are similar and the main difference is in wording.

3. The mathematical definitions of the same problem look unnecessarily different.

4. The overall approaches look similar, in particular the Figure 1 in #4706 and Figure 3 in #8776 look similar, so it is hard to tell how different the ideas behind them are.

These look suspicious because using the same names and mathematical definitions of an existing technical problem is not viewed as plagiarism. We understand that the authors' worry of using similar or the same text. Still, the extent to which the two papers are different in places they should not was more than we typically see.

The reviewers thought that the situation can be handled in a better way, as one of them commented "The papers tackle the very same problem, and in principle a comparison between them should be expected if they come from the same authors". At some point a reviewer proposed that the authors should merge the two papers in the first place. The AC went through both submissions and agreed with the reviewers.

The authors responded to the ethical concern in the rebuttal and highlighted the technical differences, which was helpful. After the rebuttal and discussions, the reviewers believe that the proposed technical approaches are sufficiently different. One thing to note is that during the rebuttal, the authors mentioned that Copyleaks gives a low score for similarity. This was not viewed as convincing, especially given that the two papers are unnecessarily different in many aspects. In fact, the AC felt that using such evidence could potentially make the paper appear more suspicious, as it could give people an impression that the papers were written to circumvent plagiarism checkers. Eventually, the recommendation was mostly based on the perceived qualities of the two submissions.

That being said, after discussions among the reviewers and among AC, SAC, and chairs, we agreed that this discussion should be conveyed to the authors, and the authors are advised to be more careful about similar issues in future submissions. As one reviewer mentioned, perhaps the authors can consider merging the two submissions to create a stronger and more thorough paper in the future.

We hope this is helpful, and thanks again for submitting to ICLR!

**Justification For Why Not Higher Score:**

The proposed methods are mildly interesting and the experiments are not as convincing as Reviewers hoped for.

**Justification For Why Not Lower Score:**

the problem is interesting and the results are solid

---

### Decision · Program_Chairs · 2024-01-16

Reject